# FBW7 couples structural integrity with functional output of primary cilia

Eleni Petsouki[1], Vasileios Gerakopoulos[1], Nicholas Szeto[2], Wenhan Chang [2], Mary Beth Humphrey [3,4] & Leonidas Tsiokas [1✉]

Structural defects in primary cilia have robust effects in diverse tissues and systems. However, how disorders of ciliary length lead to functional outcomes are unknown. We examined the functional role of a ciliary length control mechanism of FBW7-mediated destruction of NDE1, in mesenchymal stem cell (MSC) differentiation. We show that FBW7 functions as a master regulator of both negative (NDE1) and positive (TALPID3) regulators of ciliogenesis, with an overall positive net effect on primary cilia formation, MSC differentiation to osteoblasts, and bone architecture. Deletion of *Fbxw7* suppresses ciliation, Hedgehog activity, and differentiation, which are partially rescued in *Fbxw7/Nde1*-null cells. We also show that NDE1, despite suppressing ciliogenesis, promotes MSC differentiation by increasing the activity of the Hedgehog pathway by direct binding and enhancing GLI2 activity in a cilia-independent manner. We propose that FBW7 controls a protein-protein interaction network coupling ciliary structure and function, which is essential for stem cell differentiation.

[1] Department of Cell Biology, University of Oklahoma Health Sciences Center, Oklahoma City, OK, USA. [2] Department of Medicine, Division of Endocrinology and Metabolism, University of California San Francisco, San Francisco, CA, USA. [3] Department of Internal Medicine, Division of Rheumatology, Immunology, and Allergy, University of Oklahoma Health Sciences Center, Oklahoma City, OK, USA. [4] Department of Medicine, Oklahoma City Veteran's Affairs Medical Center, Oklahoma City, OK, USA. ✉email: ltsiokas@ouhsc.edu

The primary cilium is a solitary, antenna-like organelle protruding into the extracellular space[1]. It is present in virtually all cell types of the human body functioning as a signaling center for receptor tyrosine kinases, G protein coupled receptors, the Hedgehog, Notch, and Wnt pathways[2–6]. Numerous disease have been attributed to structural changes in cilia[7,8] and it is thought that changes in ciliary length may influence the output of ciliary signaling pathways[9–15]. However, the mechanisms underlying this coordination of primary cilia structure and function are complex and not well understood.

One of the signaling pathways that requires intact primary cilia for adequate activity is the Hedgehog pathway[16,17]. In the absence of the Hedgehog ligand, GLI2 and GLI3 transcription factors are proteolytically processed at the base of the primary cilium to generate transcriptional repressors, GLI2R and GLI3R[17–19]. In the presence of Hedgehog ligand, proteolytic cleavage is inhibited allowing the accumulation of full length GLI2 and GLI3 (GLI2A or GLI3A) to activate transcription of target genes[5,20]. Although all details of pathway activation have not been completely defined, it is well-established that the primary cilium provides the structural framework for maximal Hedgehog activity in vertebrates. Given the fundamental roles of the Hedgehog pathway in multiple systems, including the skeletal system and especially mesenchymal stem cell differentiation to osteoblasts and/or chondrocytes[21], we reasoned that functionality of this pathway can be used in the context of stem cell differentiation as a physiological read-out to study ciliary structure–function relationships.

Bone marrow derived MSCs are a progenitor cell source that has the ability to differentiate into multiple cell lineages, including osteoblasts, adipocytes, and chondrocytes[22–24]. Defects in MSC differentiation can have widespread effects ranging from osteopenia and chondrodysplasias to obesity and cancer metastasis. MSCs possess a primary cilium and previous work has shown that it is essential for the differentiation of these cells to osteoblasts[25]. However, exact effects of primary cilia per se, ciliary length, and contribution of cilia-based signaling and their underlying mechanisms in MSC differentiation are not completely understood.

We have identified an endogenous program regulating ciliary length in terminally differentiated cells[9]. This program involves FBW7, the recognition receptor of the SCF$^{FBW7}$ E3 ubiquitin ligase. FBW7 has established roles in maintenance, self-renewal, and differentiation of several adult stem cell types including cancer initiating cells[26,27], but the contribution of primary cilia to these effects is unknown. We previously showed that FBW7 mediates its effect on ciliary length by the timely destruction of NDE1, a negative regulator of ciliogenesis[28].

In the present study, we show that deletion of Fbxw7 in MSCs suppresses osteoblast differentiation but promotes adipocyte differentiation. Consistently, postnatal deletion of Fbxw7 in mice results in reduced bone formation and mild osteopenia in 3-month-old mice, due to reduced osteoblastogenesis. Further mechanistic experiments lead to the identification of FBW7 as a master regulator of ciliogenesis by mediating the proteasomal degradation of not only NDE1, but also TALPID3, a positive regulator of ciliogenesis, which accumulates in double Fbxw7/ Nde1- null cells and partially rescues osteoblast differentiation. TALPID3 is present at the mother centriole and positively regulates the Hedgehog pathway[29–32]. Despite its negative effect on ciliogenesis, NDE1 physically interacts with GLI2 and increases the transcriptional activity of the Hedgehog pathway in a cilium-independent mechanism. These data are consistent with a model in which FBW7 controls the abundance of positive and negative regulators of ciliogenesis impacting on Hedgehog pathway activity, and thus mediating structure–function relationship in primary cilia. These relationships are essential for stem cell differentiation.

## Results

**Postnatal deletion of Fbxw7 leads to changes in bone architecture.** The role of Fbxw7 in osteoblast differentiation in vivo has not been determined. Because of embryonic lethality of Fbxw7-null mice[33], we used the tamoxifen-inducible UbcCre$^{ERT2}$ driver to delete Fbxw7 later, in postnatal mice and examined bone architecture of the distal femur in 12-week-old male mice using microcomputed tomography (μCT). Bone volume fraction (BV/TV), connectivity density and trabecular number (Tb.N), were significantly decreased (Fig. 1a–d and Supplementary Data 1), whereas trabecular separation (Tb.Sp), a parameter that reflects overall space between trabeculae, was increased (Fig. 1e). No difference between wild type and mutant mice was detected in trabecular thickness (Tb.Th) (Fig. 1f and Supplementary Data 1). No difference was detected in cortical bone BV/TV (Fig. 1g). Serum bone-specific alkaline phosphatase levels (BALP) (Fig. 1h and Supplementary Data 1), but not Tartrate resistant acid phosphatase (TRAcP) (Fig. 1i and Supplementary Data 1), were reduced in mutant mice indicating that reduced bone mass may have been primarily caused by reduced osteoblast differentiation and/or function.

**Deletion of Fbxw7 reduces cilium incidence in primary MSCs or MSC-like cell types.** Ex vivo cultures of MSCs derived from the bone marrow of UbcCre$^{ERT2}$;Fbxw7$^{f/f}$ mice were used to test whether deletion of Fbxw7 can affect cilium incidence (percent of ciliated cells) and differentiation of MSCs to osteoblasts. We specifically examined CD106+ MSCs, called skeletal stem cells (SSCs), which give rise to osteoblasts, adipocytes, and chondrocytes[34–37]. Ex vivo deletion of Fbxw7 via 4-hyrdroxytamoxifen (4-OHT) treatment in serum starved primary cultures of mouse SSCs (CD106$^+$ MSCs) resulted in the reduction of the percentage of ciliated cells compared to untreated cells (Fig. 2a, b). Because MSC cultures can be highly heterogeneous, we also examined the effect of FBW7 depletion using RNAi in transiently transfected C3H10T1/2 cells, an MSC-like stem cell line[38,39]. Depletion of FBW7 led not only to a reduction in the percentage of ciliated cells, but also a reduction in ciliary length in remaining ciliated cells (Supplementary Fig. 1 and Supplementary Data 1). Reduced levels of cilium incidence and length were also observed in mouse embryonic fibroblasts (MEFs) transiently transfected with an Fbxw7-specific sgRNA construct (Fig. 2c–e and Supplementary Data 1). These data were consistent with our previous results in hTERT-RPE1 cells[9], that FBW7 has an essential role in ciliary length control.

**Fbxw7 deletion suppresses differentiation of MSCs to osteoblasts.** Primary cultures of UbcCre$^{ERT2}$; Fbxw7$^{fl/fl}$ MSCs allowed us to delete Fbxw7 at either a stem cell stage by inducing deletion before treatment with differentiation media (Fig. 3a), or at a stage where cells had already been committed to differentiate, by inducing deletion after addition of differentiation media (Supplementary Fig. 2 and Supplementary Data 1). Deletion of Fbxw7 in MSCs before induction of differentiation resulted in significant suppression of osteoblast differentiation (Fig. 3b), as demonstrated by Alizarin red S staining. Cell density of MSC cultures just before treatment with carrier (mock) or 4OHT were similar (Supplementary Fig. 3a), excluding effects of cell plating on differentiation. Vehicle (DMSO, mock) or 4OHT treatment in cells derived from Fbxw7$^{f/f}$, but no Cre was without effect (Supplementary Fig. 3b). In addition to Alizarin red S staining, mRNA levels of four out of five osteogenic markers were significantly

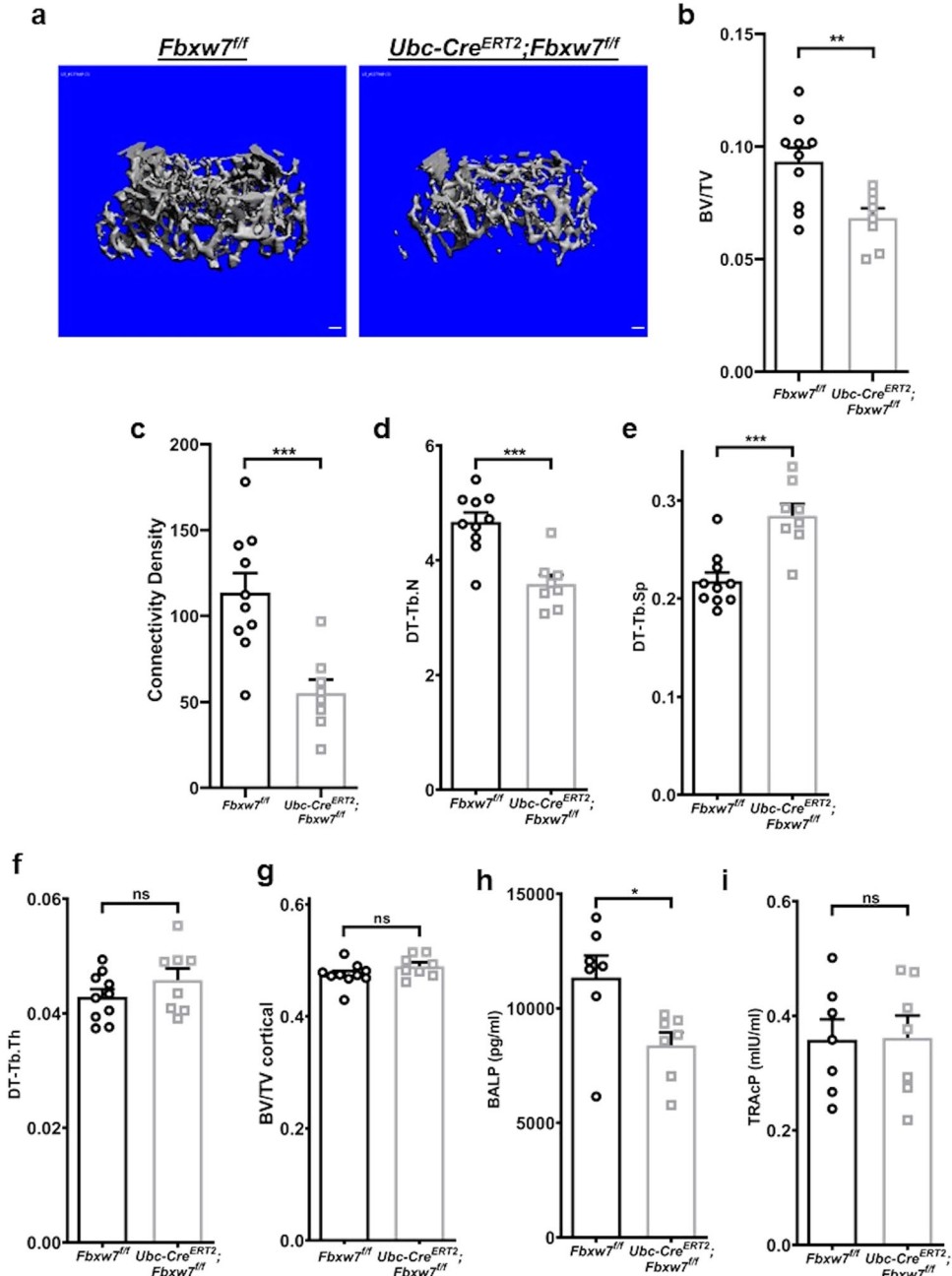

**Fig. 1 Postnatal deletion of *Fbxw7* leads to changes in bone architecture. a** Representative three-dimensional reconstitution analysis of the architecture of metaphyseal trabecular bone in the distal femur of 12-week-old *Fbxw7^{f/f}* or *UbcCreERT2; Fbxw7^{f/f}* mice treated with 4-Hydroxy-tamoxifen (4OHT). Scale bars: 100 μm. **b–g** Quantitative μCT analysis of trabecular (**b–f**) or cortical (**g**) skeletal parameters: BV/TV (Bone volume fraction) (**b**), Connectivity density (**c**), DT-Tb.N (Trabecular number) (**d**), DT-Tb.Sp (Trabecular separation) (**e**), DT-Tb.Th (Trabecular thickness) (**f**) of the metaphyseal trabecular bone or BV/TV of the cortical bone (**g**) in the distal femur of 12-week-old *Fbxw7^{f/f}* (*n* = 10) or *UbcCre^{ERT2};Fbxw7^{f/f}* (*n* = 8) mice treated with 4OHT. Data are presented as means ± SEM. Student's *t*-test, \*\**p* < 0.01, \*\*\**p* < 0.001. **h, i** ELISA analysis for bone formation (Mouse Bone-specific alkaline phosphatase, BALP) (**h**) or bone resorption (Mouse Tartrate-resistant acid phosphatase 5b, TRAcP-5b) (**i**) markers in the serum of 12-week-old *Fbxw7^{f/f}* (*n* = 7) or *UbcCreERT2; Fbxw7^{f/f}* (*n* = 7) mice treated with 4OHT. Data are presented as means ± SEM. Student's *t*-test, \**p* < 0.05.

reduced at all time-points after osteogenic induction (Fig. 3c–e and Supplementary Data 1). Despite the reduced osteoblast differentiation when *Fbxw7* deletion preceded addition of osteogenic media, deletion of *Fbxw7* after osteogenic induction resulted in a milder suppression of differentiation (Supplementary Fig. 2 and Supplementary Data 1). Efficient deletion of *Fbxw7* occurred 48 h after 4OHT addition (Supplementary Fig. 3c). In sum, these results suggested that FBW7 promoted osteoblast differentiation by acting predominantly at the stage of osteoblast lineage

commitment rather than by supporting maintenance of already committed pre-osteoblasts.

In contrast to what was seen in osteoblasts, deletion of *Fbxw7* before adipogenic treatment promoted adipogenic differentiation of MSCs isolated from *UbcCreERT2;Fbxw7^{fl/fl}* mice (Supplementary Fig. 4 and Supplementary Data 1). However, deletion of *Fbxw7* after commitment to adipogenesis showed a trend to increase differentiation without reaching statistical significance (Supplementary Fig. 4 and Supplementary Data 1). These results

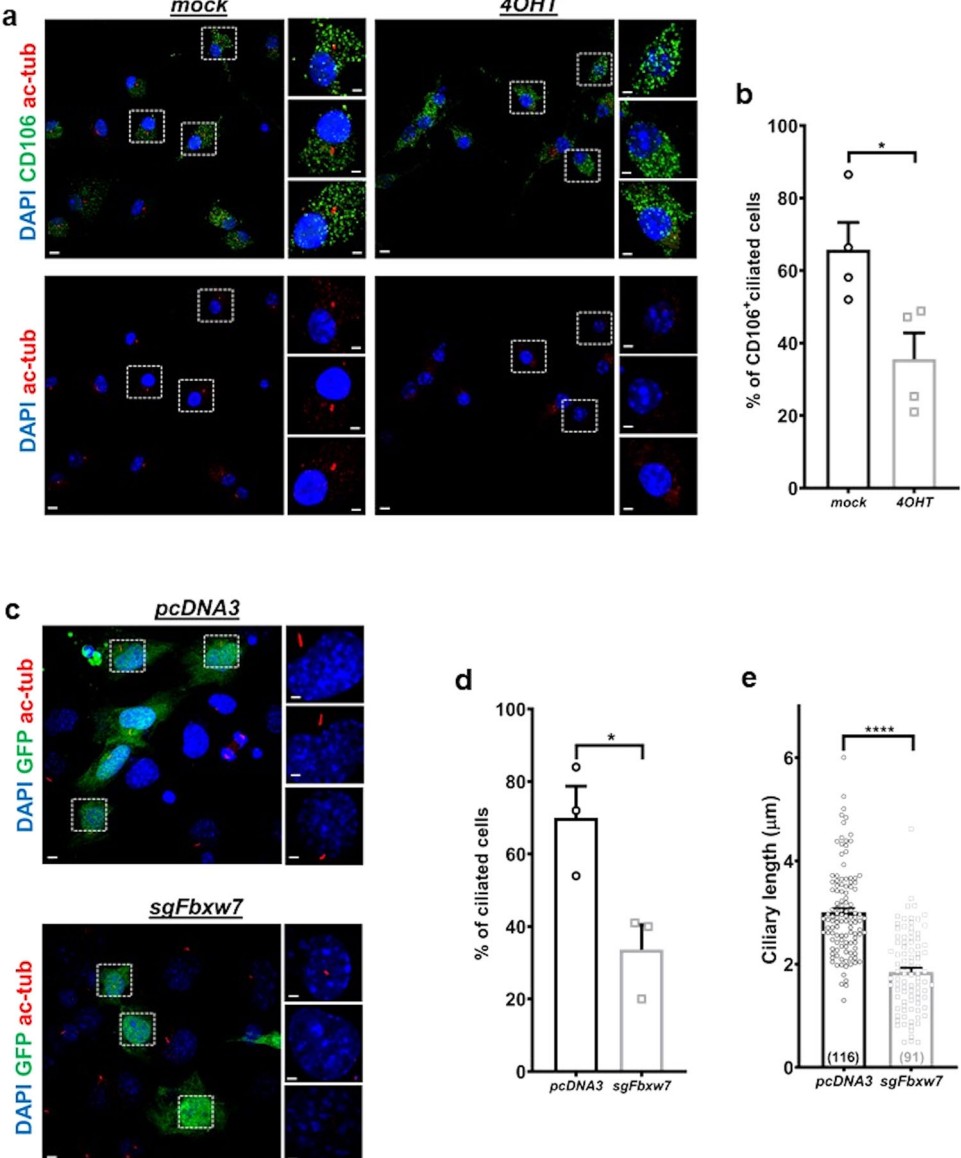

**Fig. 2 Deletion of *Fbxw7* reduces cilium incidence in MSCs and MEFs. a**, **b** Representative images of MSCs isolated from the bone marrow of *UbcCre^ERT2*;
*Fbxw7^f/f* mice, ex vivo treated with ethanol (mock) or 4OHT in ethanol, serum starved for 24 h and stained for the SSC marker CD106 (green) and primary
cilia (acetylated α-tubulin, red) (**a**). Percent of CD106+ cells with cilia in mock or 4OHT treated cells isolated from four different *UbcCre^ERT2*; *Fbxw7^f/f* mice
at 4 weeks of age (**b**). 100 mock-treated and 100 4OHT-treated CD106+ SSCs were analyzed per mouse. Scale bars: 5 μm. Scale bars in insets: 2 μm. Data
are presented as means ± SEM. Student's *t*-test, *p < 0.05. **c**–**e** Representative images of mouse embryonic fibroblasts (MEFs) transfected with GFP and
pcDNA3 or *Fbxw7*-specific sgRNA (sg*Fbxw7*), serum starved for 24 h and stained for primary cilia (acetylated α-tubulin, red) (**c**). Percent of cilium
incidence (**d**) and ciliary length (**e**) in MEFs transfected with the indicated constructs. ~100 GFP + MEFs were analyzed for each group per experiment
(n = 3) in **d**. For ciliary length analysis, the number of cells analyzed is indicated at the bottom of each bar in **e**. Scale bars: 5 μm. Scale bars in insets: 2 μm.
Data are presented as means ± SEM. Student's *t*-test, *p < 0.05.

indicated that deletion of *Fbxw7* did not generally impair the
ability of MSCs to differentiate, but it rather skewed differentia-
tion towards the adipogenic lineage at the expense of osteoblastic
differentiation. This type of effect suggested that FBW7
contributes to stem cell lineage determination rather than
affecting committed cell types.

**FBW7 regulates ciliary length through NDE1 in the C3H10T1/
2 stem cell line**. Next, we tested whether deletion of both *Nde1*
and *Fbxw7* would rescue reduced cilium incidence and ciliary
length induced by loss of *Fbxw7* alone, as previously shown in
hTERT-RPE1 cells[9]. This hypothesis was tested in the cell line
C3H10T1/2 for several reasons. First, these cells are multipotent

and have been extensively used as a surrogate cell culture system
to model osteoblast, adipocyte, or chondrocyte differentiation
in vitro[38,39]. Second, they express FBW7 and NDE1, but not
NDEL1 (Supplementary Fig. 5a). NDEL1 is a homolog of NDE1
that functions redundantly to NDE1 in ciliogenesis[40,41]. This
effect has been described in vivo and is believed to account for the
much milder phenotype of *Nde1*-null compared to *Ndel1*-null
mice restricted only to neuronal cell types[42]. Finally, the
Hedgehog pathway drives osteoblast differentiation of C3H10T1/
2 cells, and therefore cilia-based signaling is functionally relevant
in these cells[43,44].
    We generated three stable lines, lacking *Fbxw7*, *Nde1*, or both
using CRISPR/Cas9 gene editing (Supplementary Fig. 5b–d).

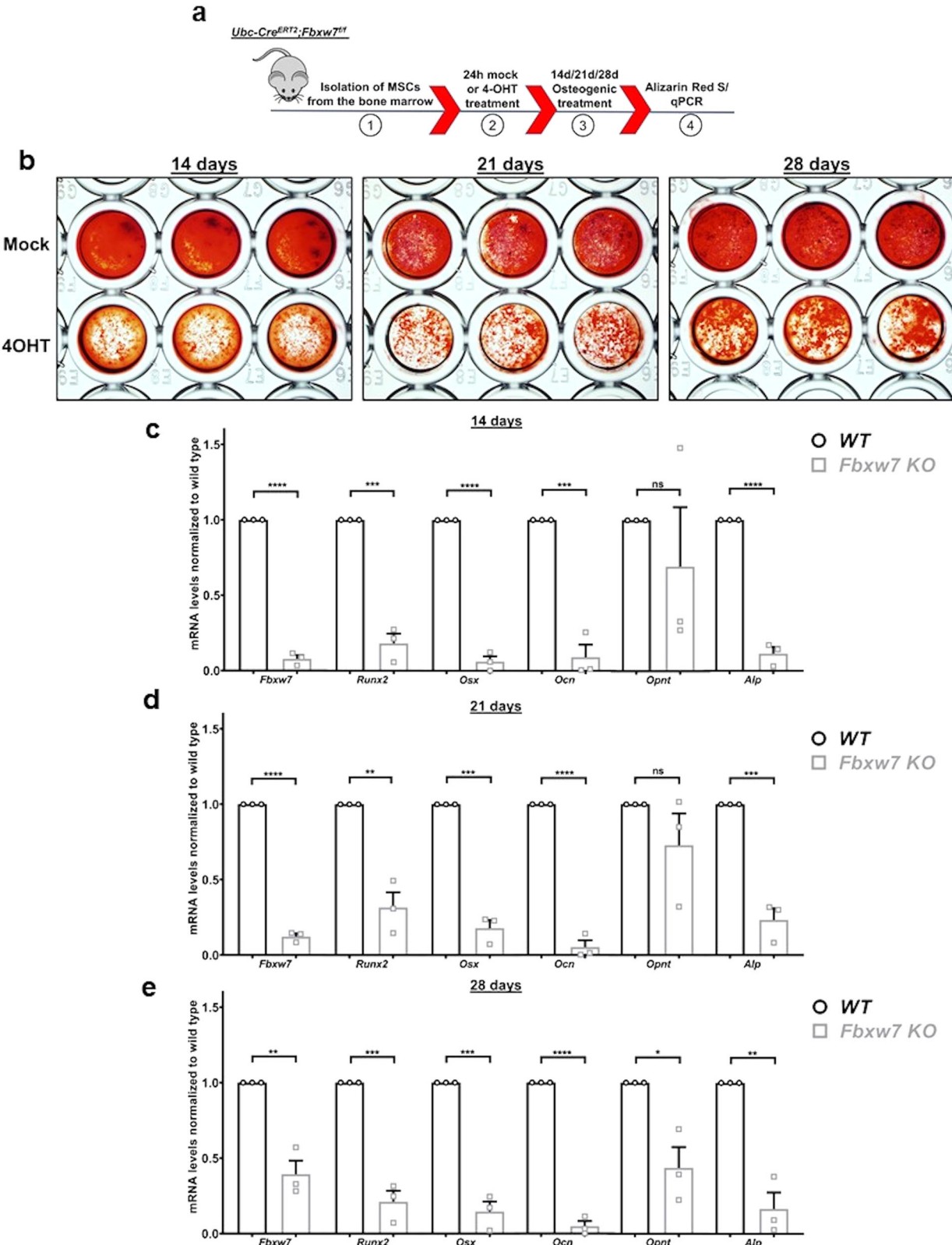

Because current batches of the NDE1 antibody recognize an additional band in mouse lysates, we used MEFs from $Nde1^{-/-}$ mice to identify the band corresponded to NDE1. The lower band detected by the NDE1 antibody corresponded to NDE1 (Supplementary Fig. 5c, d). As expected, cells lacking *Fbxw7* showed an upregulation of NDE1 (Supplementary Fig. 5e). *Fbxw7*-null cells showed a reduction in the percentage of ciliated

cells and ciliary length in remaining ciliated cells (Fig. 4 and Supplementary Data 1), as shown in SSCs (Fig. 2a, b). This effect was demonstrated via visualization of primary cilia either with acetylated tubulin (Fig. 4a–f and Supplementary Data 1) or Arl13b (Fig. 4g–l and Supplementary Data 1). Deletion of *Nde1* did not increase the percentage of ciliated cells 24 h after serum starvation, but rather ciliary length (Fig. 4f, l and Supplementary

**Fig. 3 *Fbxw7* deletion before osteogenic induction suppresses differentiation of MSCs to osteoblasts. a** Diagram showing an experimental setup for ex vivo deletion of *Fbxw7*. Isolation of MSCs from the bone marrow of *UbcCre^ERT2; Fbxw7^f/f* mice (1) was followed by 4OHT-induced deletion of *Fbxw7* (2), osteogenic induction (3) and analysis of osteoblast differentiation (4). **b**–**e** Osteoblast differentiation of *UbcCre^ERT2; Fbxw7^f/f*-derived MSCs treated with mock or 4OHT before osteogenic induction (*n* = 3 different mice). Differentiation was measured at 14, 21, and 28 days after osteogenic induction via Alizarin Red S staining (**b**) and mRNA levels of osteoblast differentiation markers *Runx2*, *Osterix* (*Osx*), *Osteocalcin* (*Ocn*), *Osteopontin* (*Opnt*) and Alkaline phosphatase (*Alp*) at 14 (**c**), 21 (**d**), and 28 days (**e**) after osteogenic induction. mRNA levels of *Fbxw7* were analyzed to confirm deletion of the gene. Images were taken from 96-well plates. Well diameter: 5 mm (**b**). Data are presented as means ± SEM. Student's *t*-test, *$p < 0.05$, **$p < 0.01$, ***$p < 0.001$, ****$p < 0.0001$.

Data 1), as we had previously shown in NIH3T3 and hTERT-RPE1 cells[28]. In double mutant cells, the percentage of ciliated cells was similar to wild type levels (Fig. 4e, k and Supplementary Data 1), while ciliary length was significantly higher than wild type cells (Fig. 4f, l and Supplementary Data 1).

**FBW7 regulates functional integrity of cilia in C3H10T1/2 cells.** While experiments above showed that the FBW7/NDE1 module is essential for cilia structure, they could not inform us about cilia function. Thus, we tested whether cilia per se are essential for osteoblast differentiation. We inhibited cilia formation by CRISPR/Cas9-mediated inactivation of the *Ift88* gene in C3H10T1/2 cells. Cells lacking cilia failed to differentiate to osteoblast-like cells, as determined by expression of alkaline phosphatase (ALP), a well-established marker of osteoblast differentiation in these and other multipotent cells (Supplementary Fig. 6). Osteoblast differentiation was severely reduced in single *Fbxw7*-deleted and *Nde1*-deleted cells compared to wild type cells, but partially restored in double mutant cells (Fig. 5a–c and Supplementary Data 1). Similar effects were seen on mRNA levels of *Runx2* and *Osterix*, two master regulators of osteoblast differentiation (Fig. 5d, e and Supplementary Data 1). In addition, we used MLN4924, an inhibitor of Cullin-dependent ligases, including FBW7[45]. Initiation of treatment of wild type C3H10T1/2 cells with MLN4924 before osteogenic induction resulted in suppression of differentiation. However, initiation of MLN4924 treatment only after osteogenic induction had no effect on differentiation (Supplementary Fig. 7 and Supplementary Data 1). These data were consistent with data using 4-OHT-inducible deletion of *Fbxw7* in MSCs (Fig. 3 and Supplementary Fig. 2), suggesting that SCF^FBW7 activity at the pre-commitment stage, but not after commitment, is more relevant to osteoblast differentiation. These results also supported the hypothesis that the FBW7-mediated degradation of NDE1 is essential for osteoblast differentiation. To test whether cilia were functional in double mutant cells, cells were transiently transfected with an *Ift88*-specific sgRNA (Fig. 5f, g). ALP levels were dramatically suppressed in these cells, suggesting that deletion of *Fbxw7* and *Nde1* led to the formation of functional cilia (Fig. 5). While the reduced level of differentiation in FBW7 depleted cells could be consistent with the reduced percentage of ciliated cells and ciliary length, the reduced level of differentiation in *Nde1*-null cells was somewhat unexpected based on effects on cilia alone and suggested that abnormally short or long cilia lead to reduced function. Similar results were obtained in additional clones of C3H10T1/2 cells lacking *Nde1*, ruling out clonal effects (Supplementary Fig. 8). Based on these data, we reasoned that while deletion of *Nde1* led to abnormally long cilia, these cilia were not functional, yet they acquire some level of functionality in double mutant cells.

Considering the central role of the Hedgehog pathway in osteoblast differentiation and the dependence of this pathway on structural integrity of primary cilia[5,20,21], we transfected all four cell lines with a Gli-reporter construct and determined basal level activity of the Hedgehog pathway. As shown in Fig. 6a, pathway activity correlated with differentiation levels. The Hedgehog

pathway has been reported to induce activation of canonical Wnt/β-catenin pathway in C3H10T1/2 cells by upregulating the expression levels of certain Wnt ligands, such as WNT9A[46]. WNT9A is expressed in osteoblasts[47] and has a positive role in canonical Wnt signaling[48–50]. Therefore, we examined activity of canonical Wnt/β-catenin pathway and *Wnt9a* mRNA levels in all four cell lines. Both paralleled activity of Hedgehog pathway (Fig. 6b, c and Supplementary Data 1). These data suggested that reduced Hedgehog activity leads to reduced canonical Wnt/β-catenin activity, which is essential for osteoblast differentiation.

The reduced activity of the Hedgehog pathway in *Fbxw7*-null or *Nde1*-null cells led us to hypothesize that functionality of cilia-related Hedgehog effectors might have been compromised. Thus, we tested whether overexpression of various forms of GLI2 or GLI3 could rescue differentiation in *Nde1*-null cells (Supplementary Fig. 9a–d). Full length or constitutively active GLI2 increased differentiation in these cells, suggesting the possible defects in GLI2-mediated signaling may account for the impaired cilia functionality in *Nde1*-null cells[20,51]. Interestingly, overexpression of a constitutively active full-length GLI3 construct (GLI3 P1–P6) had no effect in *Nde1*-null cells[20], while further suppressed differentiation in *Fbxw7*-null cells (Supplementary Fig. 9c, d), highlighting the importance and specificity of GLI2 in cilia-mediated osteoblastogenesis in this system. Here, it is important to emphasize that it was not GLI3R that could account for the negative effect of GLI3 P1–P6 on *Fbxw7*-null cells, because this GLI3 form is resistant to proteolytic cleavage. Therefore, we focused on GLI2.

**NDE1 physically interacts with, increases the transcriptional activity of GLI2 and promotes its nuclear localization.** Hedgehog activity was decreased in *Nde1*-null cells despite the presence of cilia. This prompted us to test whether NDE1 increased the activity of GLI2 and physically interacted with GLI2. Transient transfection in C3H10T1/2 and HEK293T cells indicated that GLI2 activity was increased in the presence of NDE1 (Fig. 6d, e and Supplementary Data 1) and NDE1 co-immunoprecipitated with GLI2 (Fig. 6f), but not with irrelevant protein, bacterial alkaline phosphatase (FLAG-BAP, Supplementary Fig. 10b). It should be noted that these experiments were not done under conditions that favor cilia formation and, in addition, overexpression of NDE1 has a strong effect on suppression of cilia formation (Supplementary Fig. 9e, f and Supplementary Data 1). Therefore, the positive effect of NDE1 on GLI2 activity should be mediated via a cilia-independent manner or most likely, at a step downstream of cilia.

To determine whether NDE1 affected the nuclear localization of GLI2, we transiently transfected wild type and *Nde1*-null C3H10T1/2 cells with a constitutively active form of GLI2 (GLI2ΔN) tagged with myc. This construct is resistant to proteolytic cleavage and lacks an N-terminal inhibitory domain. While in the majority of wild type cells, GLI2ΔN was localized in the nucleus, in *Nde1*-null cells it was equally distributed in the nucleus and cytoplasm (Fig. 6g). Quantification of fluorescence intensity in the nucleus versus the cytoplasm indicated much reduced accumulation of GLI2ΔN in the nucleus in *Nde1*-null

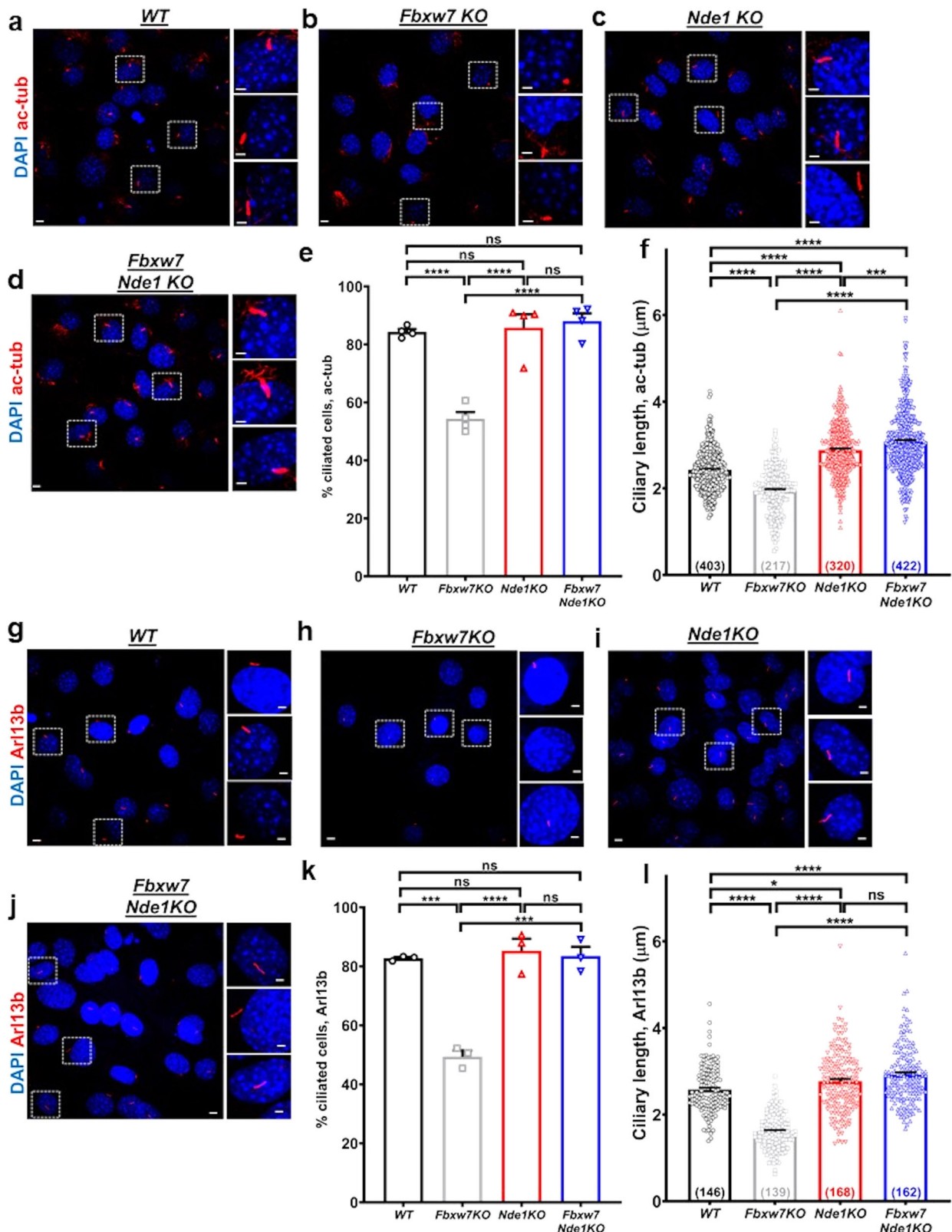

compared to wild type cells (Fig. 6h and Supplementary Data 1). These data suggest that NDE1 promotes the nuclear localization of GLI2, and could help explain the reduced overall GLI activity in *Nde1*-null cells. They further indicate that while NDE1 functions as a negative regulator of ciliogenesis, it functions as a positive regulator of cilia based signaling such as Hedgehog signaling, inversely coupling ciliary structure and function.

**Identification of TALPID3 as a possible target of FBW7.** Because osteoblast differentiation was partially rescued in double mutant cells, while cilia appeared grossly structurally similar to cilia of *Nde1*-null cells, we reasoned that positive regulator(s) of the Hedgehog pathway might be additional potential targets of FBW7 that could accumulate in double mutant cells providing some degree of rescue. Screening all known positive regulators of

**Fig. 4 FBW7 reduces cilia formation and length via NDE1 in the C3H10T1/2 stem cell line. a–d** Representative images of wild type, *Fbxw7KO, Nde1KO*, and *Fbxw7/Nde1KO* C3H10T1/2 cells after 24 h of serum starvation stained for primary cilia (acetylated α-tubulin, red). Percentages of ciliated cells (**e**) and ciliary length (**f**) in wild type, *Fbxw7KO, Nde1KO*, and *Fbxw7/Nde1KO* C3H10T1/2 cells. Hundred to hundred and fifty cells were analyzed per experiment (*n* = 3) in **e**. For ciliary length analysis, the number of cells analyzed is indicated at the bottom of each bar in **f**. Scale bars: 5 μm. Scale bars in insets: 2 μm. Data are presented as means ± SEM. One-way ANOVA with Sidak's multiple comparisons test, ***p < 0.001, ****p < 0.0001. **g–j** Representative images of wild type, *Fbxw7KO, Nde1KO*, and *Fbxw7/Nde1KO* C3H10T1/2 cells after 24 h of serum starvation stained for primary cilia (Arl13B, red). Percentages of ciliated cells (**k**) and ciliary length (**l**) in wild type, *Fbxw7KO, Nde1KO* and *Fbxw7/Nde1KO* C3H10T1/2 cells. Hundred to hundred and fifty cells were analyzed per experiment (*n* = 3) in **k**. For ciliary length analysis, the number of cells analyzed is indicated at the bottom of each bar in **f**. Scale bars: 5 μm. Scale bars in insets: 2 μm. Data are presented as means ± SEM. One-way ANOVA with Sidak's multiple comparisons test, ***p < 0.001, ****p < 0.0001.

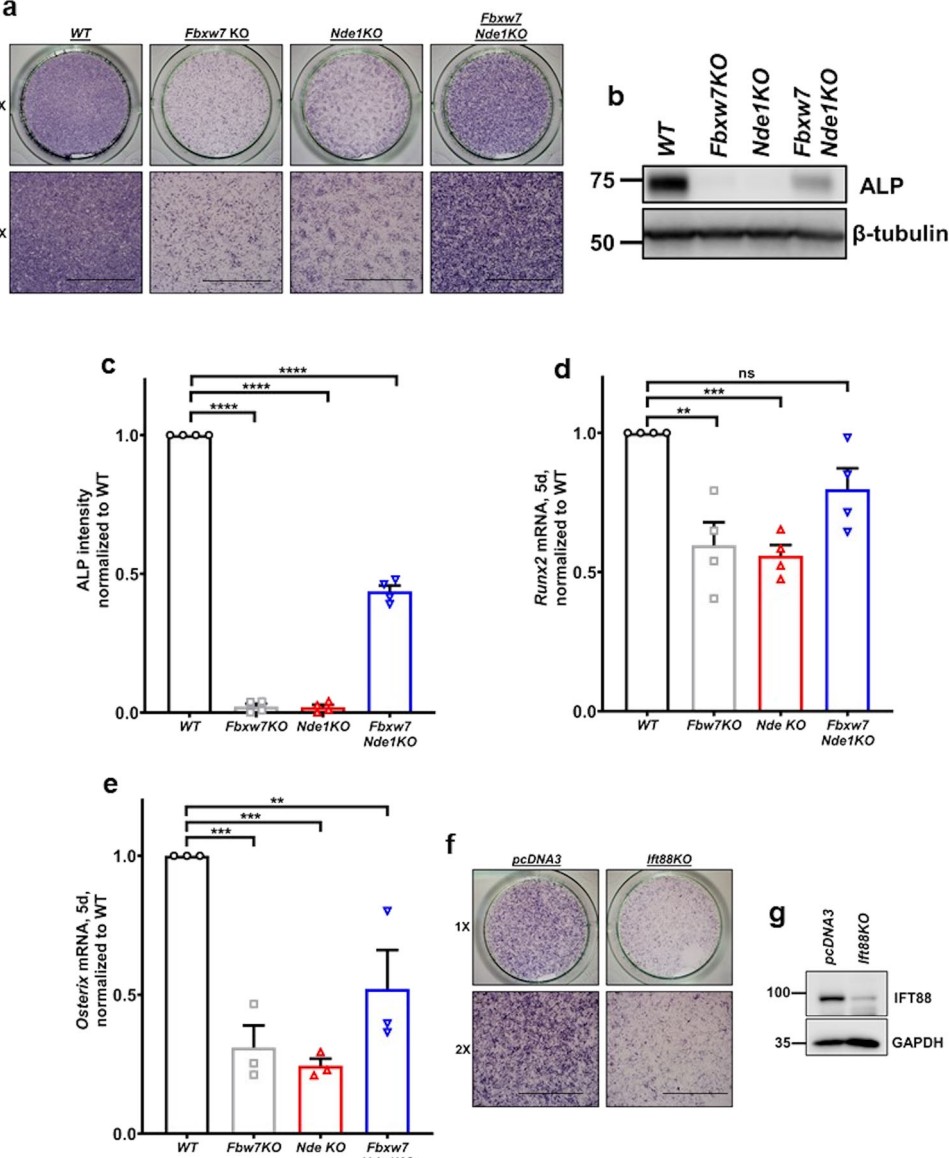

**Fig. 5 FBW7-NDE1 axis affects osteoblast differentiation via effects on primary cilia in C3H10T1/2 cells. a** Representative images for ALP staining of wild type, *Fbxw7KO, Nde1KO*, and *Fbxw7Nde1KO* C3H10T1/2 cells (*n* = 3). Images at ×1 or ×2 magnification were taken from 24-well plates. Well diameter: 16 mm. Scale bar: 5 mm. **b, c** Expression levels of ALP in indicated cells (**b**) and summary data (**c**). Data are presented as means ± SEM from four independent experiments. One-way ANOVA with Dunnett's multiple comparisons test, ****p < 0.0001. **d, e** mRNA levels of osteoblast differentiation markers *Runx2* and *Osterix* (*Osx*) in wild type, *Fbxw7KO, Nde1KO*, and *Fbxw7Nde1KO* C3H10T1/2 cells. Data are presented as means ± SEM from four or three independent experiments (**d** or **e**, respectively). One-way ANOVA with Dunnett's multiple comparisons test, **p < 0.01, ***p < 0.001. **f, g** ALP staining after transfection of *Fbxw7Nde1KO* C3H10T1/2 cells with pcDNA3 or an *Ift88*-specific sgRNA. Images at ×1 or ×2 magnification were taken from 24-well plates. Well diameter: 16 mm (**f**). Scale bar: 5 mm. Expression levels of IFT88 in *Fbxw7Nde1KO* C3H10T1/2 cells transfected with the indicated constructs (**g**).

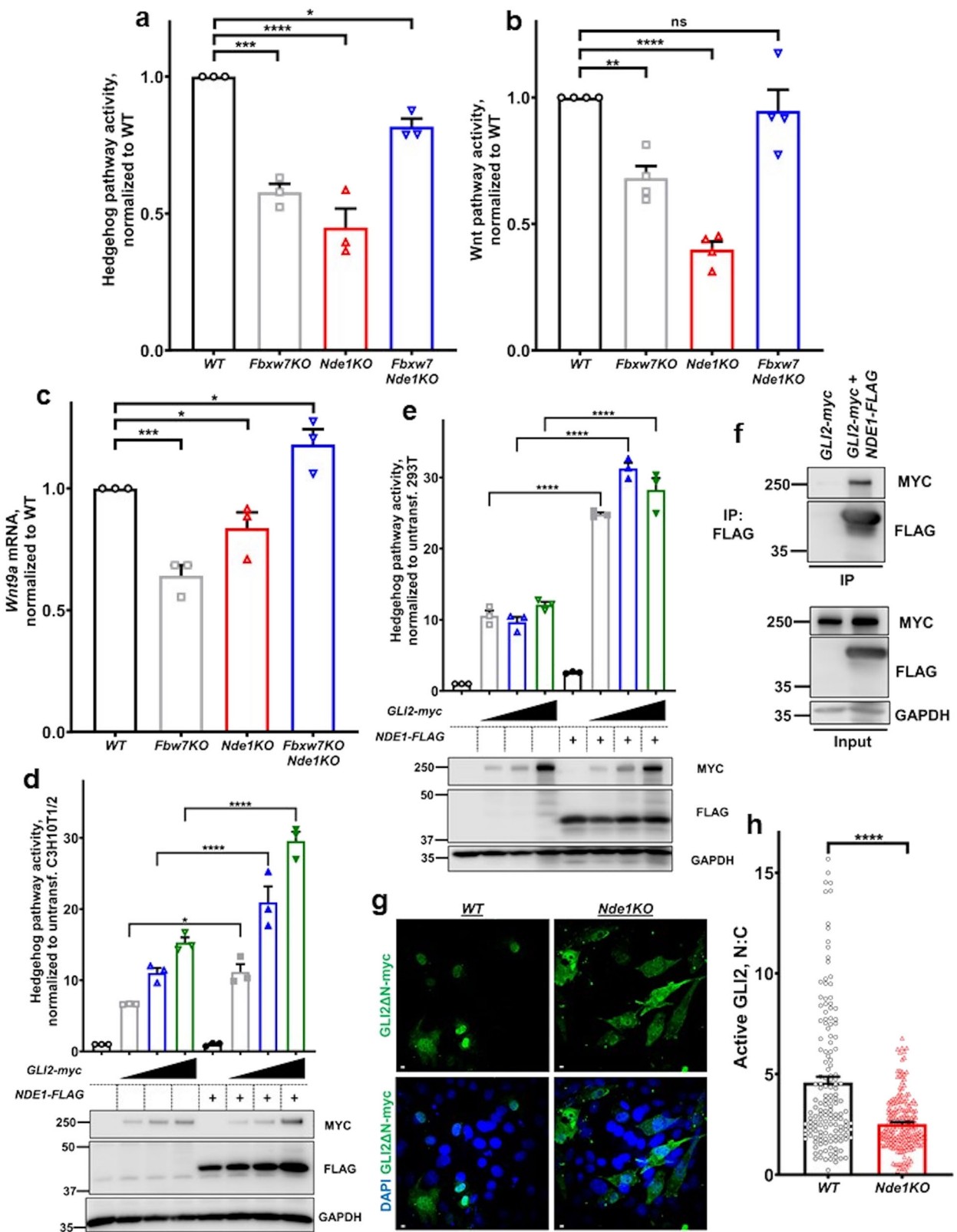

the Hedgehog pathway for the presence of optimal FBW7 phosphodegrons, we identified Fused and TALPID3 as potential targets. We proceeded with TALPID3, because it had four optimal FBW7 phosphodegrons (Fig. 7d). In addition, TALPID3 is required for early stages of cilia formation and organization of transition fibers to assemble the ciliary gate at the base of the cilium[30], which would be consistent with the small but significant

increase in ciliary length in double mutant cells (Fig. 4f, l and Supplementary Data 1). As predicted, all FBW7 isoforms co-immunoprecipitated with TALPID3, albeit at different levels (Fig. 7a, b). TALPID3 did not interact with an irrelevant protein such as KLF4 (Supplementary Fig. 10). Ubiquitinylation experiments in HEK293T cells suggested that co-expression of TAL-PID3 with wild type, but not ubiquitin ligase dead FBW7

**Fig. 6 Hedgehog pathway activity in single and double mutant cells and its modulation by NDE1. a** Summary data of Hedgehog pathway activity normalized to wild type, after transfection of wild type, *Fbxw7KO*, *Nde1KO*, and *Fbxw7Nde1KO* C3H10T1/2 cells with a Gli-reporter construct. Data are presented as means ± SEM from three independent experiments. One-way ANOVA with Dunnett's multiple comparisons test, *$p < 0.05$, ***$p < 0.001$, ****$p < 0.0001$. **b** Summary data of Wnt pathway activity normalized to wild type, after transfection of wild type, *Fbxw7KO*, *Nde1KO*, and *Fbxw7Nde1KO* C3H10T1/2 cells with a βcatenin-reporter construct. Data are presented as means ± SEM from four independent experiments. One-way ANOVA with Dunnett's multiple comparisons test, **$p < 0.01$, ****$p < 0.0001$. **c** mRNA levels of *Wnt9a* in wild type, *Fbxw7KO*, *Nde1KO*, and *Fbxw7Nde1KO* C3H10T1/2 cells ($n = 3$ experiments). Data are presented as means ± SEM. One-way ANOVA with Dunnett's multiple comparisons test, *$p < 0.05$, ***$p < 0.001$. **d, e** Summary data of Hedgehog pathway activity, normalized to untransfected, after transfection of C3H10T1/2 (**d**) or HEK293T cells (**e**) with a Gli-reporter construct and the indicated constructs ($n = 3$ experiments). Expression of the indicated constructs is shown at the bottom of each graph. Data are presented as means ± SEM. One-way ANOVA with Sidak's multiple comparisons test, *$p < 0.05$, ****$p < 0.0001$. **f** Physical interaction of GLI2-myc and NDE1-FLAG constructs in HEK293T cells. **g, h** Representative images of wild type and *Nde1KO* C3H10T1/2 cells transiently transfected with a GLI2ΔN-myc construct encoding an uncleavable form of GLI2 (active GLI2). Cells were stained for myc-tag (green, **g**). Summary data from of the nuclear:cytosol (N:C) ratio of GLI2ΔN-myc in the indicated cells (**h**). Data were pooled from three independent experiments and presented as means ± SEM. Student's *t*-test, ****$p < 0.0001$.

mutants, resulted in increased ubiquitinylation of TALPID3 (Fig. 7c). Interestingly, an N-terminally truncated form of FBW7 that has been widely used for functional studies was stabilized when co-transfected with TALPID3, suggesting that binding of TALPID3 to FBW7 may have blocked its auto-ubiquitylation[52] supporting the idea that TALPID3 is a genuine target of FBW7 (Fig. 7a). Consistent with the positive effect of TALPID3 on Hedgehog signaling and osteoblast differentiation, transient depletion of *Talpid3* mRNA in wild type or double mutant *Fbxw7/Nde1* cells, resulted in marked reductions of both functional read-outs (Fig. 7e–j and Supplementary Data 1). Efficient depletion of mouse TALIPD3 by *Talpid3* siRNA was confirmed in transfected C3H10T1/3 cells (Supplementary Fig. 10c, d). These data suggest FBW7 controls the abundance of TALPID3 in addition to NDE1 and the coordinated effect of these, and possibly other yet unidentified proteins can contribute to the cilia structure–function coupling in cells lacking *Fbxw7*.

## Discussion

While it is known that abnormally long or short cilia can result in severe phenotypes in diverse tissues and systems, it is not clear how changes in ciliary length can influence signaling output resulting in these robust phenotypes. We have shown earlier that the FBW7-mediated proteasomal degradation of NDE1 functions as a rheostat for ciliary length[9]. Results from our present study suggest that ciliary signaling output is tuned to ciliary length by direct physical interactions of proteins, that mediate structural roles with proteins mediating functional roles (Fig. 8). This conclusion is based on the following lines of evidence. First, FBW7 controls the abundance of both positive and negative regulators of ciliogenesis, TALPID3 and NDE1, respectively. Second, NDE1 physically interacts with GLI2 increasing its nuclear localization and transcriptional activity. Third, the FBW7/NDE1/TALPID3/GLI2 network of protein–protein interactions and activities is essential for both cilia formation and differentiation of MSCs to osteoblasts and postnatal deletion of *Fbxw7* leads to reduced bone mass and osteoblast activity. While the differentiation of MSCs to osteoblast was utilized as a model system to decipher molecular and cellular mechanisms of cilia structure and function, our results have implications in diseases associated with bone loss in aging and in the pathophysiology of not only *bona-fide* ciliopathies, such as Joubert syndrome (TALPID3)[53], but also of secondary ciliopathies such as cancer (FBW7)[54,55] and microcephaly (NDE1)[41].

Our approach to mechanistically dissect out structure–function relationships of primary cilia entailed the identification of a functional assay for the primary cilium in cell culture that would also bear physiological relevance in the whole organism. Our data showed that deletion of *Fbxw7* in primary MSCs and C3H10T1/2

cells suppressed both cilium incidence and osteoblast differentiation and promoted adipocyte differentiation. Consistently, postnatal deletion of *Fbxw7* in mice reduced bone mass and levels of BALP, a marker of osteoblastic activity in the serum. Double deletion of *Fbxw7* and *Nde1* in C3H10T1/2 cells rescued cilia formation and osteoblast differentiation. Triple deletion of *Fbxw7*, *Nde1*, and *Ift88* eliminated rescued cilia and suppressed differentiation. These data lead us to propose that the FBW7/NDE1 module had a functional role in the differentiation of MSCs to osteoblasts, and such a role was mediated at least in part, via an effect on cilia. However, we unexpectedly found that cells lacking *Nde1* showed a reduction in differentiation despite the fact that cilia were present in these cells. These results indicated that restoring ciliary length does not automatically mean that cilia function is also restored. Further mechanistic experiments led to two key observations: (1) TALPID3, a positive regulator of ciliogenesis and Hedgehog activity, is a target of FBW7 and (2) NDE1 is a positive regulator of GLI2 and a negative regulator of ciliary length. These data lead us to propose a model in which, FBW7 controls structural integrity of cilia by regulating the abundance of NDE1 and TALPID3. Other yet unidentified candidates also could be playing important roles. Upregulation of NDE1 in *Fbxw7*-null cells had a dominant effect on ciliary length over TALPID3, leading to an overall reduced number of cells with cilia and a reduction of ciliary length in remaining ciliated cells. As reduced overall ciliation impacts on Hedgehog activity limiting its activity, it is reasonable to propose that reduced number of ciliated cells and/or length suppressed differentiation in *Fbxw7*-null cells. In *Nde1*-null cells however, although ciliary length is increased, overall Hedgehog activity and specifically, GLI2 activity is reduced. Overexpression of GLI2, but not GLI3 in *Nde1*-null cells rescued differentiation. Furthermore, overexpression of a constitutively mutant of GLI2 (GLI2ΔN) did not localize to the nucleus in *Nde1*-null cells as efficiently as in wild type cells, suggesting a possible role of NDE1 in facilitating nuclear translocation of activated GLI2. In double mutant cells, the partial rescue in Hedgehog activity could be attributed to an increase in the pool of activated GLI2 triggered by a possible stabilization of TALPID3 levels. The possibility of TALPID3 promoting the nuclear localization of GLI2 and GLI3 has been suggested[31] and whether such a role is due to a non-ciliogenic function is a subject for future studies. Regardless, the accumulation of TALPID3 in double *Fbxw7*-null/*Nde1*-null cells could further promote ciliogenesis in the absence of NDE1 and increase Hedgehog activity directly, as has been shown previously by others[31] or indirectly. Functional contribution of TALPID3 in Hedgehog activity and osteoblast differentiation in double *Fbxw7/Nde1*-null cells was shown. This model is consistent with our data unraveling an interplay between cilia structure and cilia function.

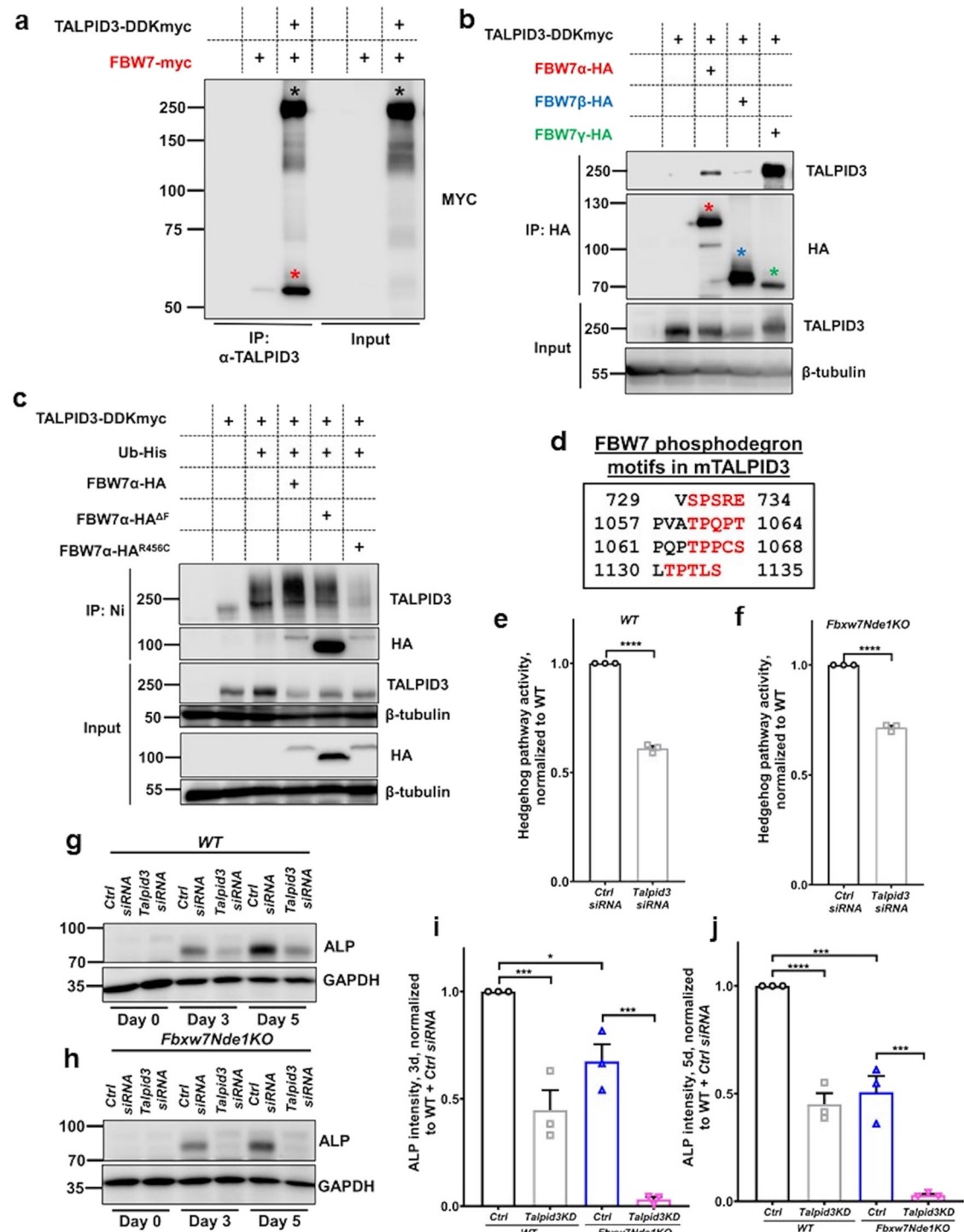

**Fig. 7 Identification of TALPID3 as a target of FBW7. a**, **b** Physical interaction of TALPID3-myc with the indicated constructs of FBW7 in HEK293T cells. Black and red asterisks in **a** indicate TALPID3-myc and FBW7-myc, respectively. Red, blue and green asterisks in **b** indicate FBW7α, FBW7β, and FBW7γ, respectively. **c** Ubiquitinylation assay of TALPID3-myc in HEK293T cells after co-transfection with 6-His-tagged Ubiquitin and the indicated HA-tagged FBW7 constructs. **d** Predicted FBW7 phosphodegron motifs in mouse TALPID3. **e**, **f** Summary data of Hedgehog pathway activity normalized to wild type after transfection of wild type (**e**) and *Fbxw7Nde1KO* (**f**) C3H10T1/2 cells with a Gli-reporter construct and *Talpid3* siRNA or control siRNA ($n = 3$ experiments). Data are presented as means ± SEM. Student's *t*-test, ****p < 0.01*, *****p < 0.0001*. **g–j** Expression levels of ALP in wild type (**g**) or *Fbxw7Nde1KO* (**h**) C3H10T1/2 cells, transfected with control siRNA or *Talpid3* siRNA, at 0 days, 3 days, and 5 days of osteogenic differentiation. Summary data of **g** and **h**, normalized to wild type cells transfected with mock siRNA, at 3 days (**i**) or 5 days (**j**) of osteogenic differentiation ($n = 3$ experiments). Data are presented as means ± SEM. One-way ANOVA with Sidak's multiple comparisons test, **p < 0.05*, ****p < 0.001*, *****p < 0.0001*.

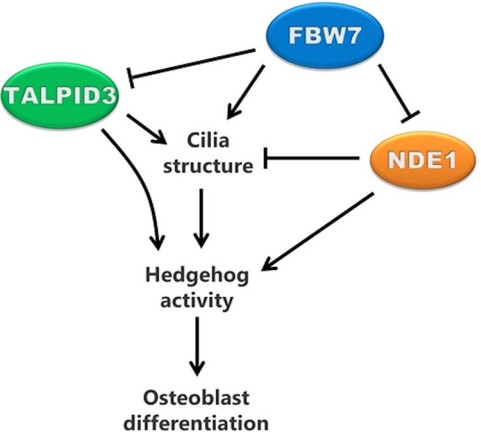

**Fig. 8 Working model.** Working model depicting the role of the FBW7–NDE1–TALPID3 pathway in primary cilia structure, Hedgehog pathway activity, and osteoblast differentiation.

Our data show that mechanisms mediating ciliary length are relevant in progenitor/stem cell differentiation and lineage commitment. We show that time-dependent deletions of *Fbxw7* before or after commitment to osteoblast or adipocyte differentiation act primarily at the stage, which stem cells commit to a certain lineage. We also show that in *Fbxw7*-null cells, both Hedgehog and canonical Wnt/β-catenin activities are reduced, which are restored in double mutant cells. Previous work has shown that Hedgehog is upstream of canonical Wnt activity in osteoblast differentiation[46], and both pathways promote osteoblastogenesis and inhibit adipogenesis[56,57]. Our data are consistent with these findings. The enhanced adipogenesis at the expense of osteoblastogenesis in *Fbxw7*-null MSCs indicate that FBW7 may protect from osteopenia during aging, as reduced bone mass associated with an abnormal accumulation of adipocytes on bones is a hallmark of aging-induced osteopenia[58].

FBW7 is one of the most commonly mutated tumor suppressors. Based on our current and previous data showing that depletion of FBW7 affects structure and function of primary cilia[9], it is tempting to speculate that part of the tumorigenic program induced upon the loss of functional FBW7 could be attributed to cilia and cilia-based signaling. We envision that loss of functional FBW7 could promote tumorigenicity in GLI2-dependent tumors or tumors that arise independently of Hedgehog signaling, and they do not require an intact cilium for initiation and/or progression. However, loss or inactivation of *Fbxw7* could have an antitumor or protective effect in Hedgehog-dependent tumors including tumors dependent on inactivating mutations in *PTCH1* or activating mutations in *SMO*, which both require an intact cilium for tumor growth. Inactivating mutations in *PTCH1* and activating mutations in *SMO* have been detected in 73% or 20%, respectively, of patients with basal cell carcinoma (BCC), a stem cell-based malignancy accounting for ~90% of all solid tumors. Five percent of BCC patients in this study also had inactivating mutations in *FBXW7*[59]. It would be interesting to know whether tumor aggressiveness correlated with the presence or absence of *PTCH1* or *SMO* with *FBXW7* mutations. According to our model, patients with both mutations (*PTCH* and *FBXW7* or *SMO* and *FBW7*) should have better prognosis.

*NDE1* is mutated in patients with primary microcephaly. We had proposed earlier that loss of NDE1 in radial glial progenitors may have caused a delay in G1 to S transition due to abnormally long cilia[28], which was later supported by in vivo experiments in rats[41]. Our current data raise the possibility that reduced Hedgehog activity could also contribute to the delay in cell cycle

progression in this context, leading to premature cell cycle exit and depletion or reduction of the radial glial progenitor pool size. Interestingly, bi-allelic loss-of-function variations in *SMO* or Hedgehog acyl-transferase (*HHAT*) genes in humans led to many developmental anomalies including microcephaly[60,61]. Our data support the idea that microcephaly in humans caused by inactivating mutations in *NDE1* might be due to abnormally long cilia and/or reduced Hedgehog activity (GLI2).

Joubert syndrome represents a genetically and phenotypically heterogeneous group of disorders characterized by hypoplasia of the cerebellar vermis, skeletal abnormalities, retinal dystrophy, and renal anomalies. TALPID3 is mutated in patients with Joubert syndrome 23, which specifically show hydrocephaly, short-rib thoracic dysplasia 14 with polydactyly, and other skeletal malformations[53]. Our data that TALPID3 is required for osteoblastogenesis and possibly, chondrogenesis can be consistent with human data. From a cilia-centric point of view however, it appears that Joubert syndrome 23 and microcephaly may be mediated by diametrically opposite mechanisms, as Joubert Syndrome 23 is caused by loss of cilia, while microcephaly may be caused by abnormally long cilia. Considering that both NDE1 and TALPID3 have a positive effect on Hedgehog signaling, both diseases seem to share a common defect to maintain a certain level of Hedgehog activity despite cilia being structurally very different.

Overall, our study identified a protein–protein interaction network with an essential role in coupling cilia structure with function. Although the studies were performed in a model system of stem cell differentiation, which is highly amenable to in vitro, ex vivo, and in vivo approaches of cell differentiation, and in which primary cilia have an established role, mechanistic information was generated that could help understand cilia function not only in normal conditions, such as stem cell differentiation and aging, but also in diverse diseases such as cancer, microcephaly, and Joubert syndrome.

## Methods

**Mice**. Mice were maintained under pathogen-free conditions in the barrier facility of The University of Oklahoma Health Sciences Center. All procedures were approved by the Institutional Care and Use Committee of University of Oklahoma Health Sciences Center.

**Microcomputed tomography (μCT) analysis**. *UbcCre^ERT2; Fbxw7^f/f* male mice were crossed with *Fbxw7^f/f* female mice (all mice were on C57Bl/6 background). To induce deletion of *Fbxw7*, nursing dams were intraperitoneally injected with 4-Hydroxy-Tamoxifen from postnatal days 2 to 6 (P2–P6)[62]. Male 12-weeks-old mice were sacrificed, soft tissue was removed and femur was analyzed at the UCSF CCMBM Skeletal Biology and Biomechanics Core for changes in 3-dimension structural parameters.

**ELISA**. Serum was obtained from 12-week-old mice and analyzed by ELISA for Mouse Bone Specific alkaline phosphatase (BALP, MyBiosource, #MBS281206) and Mouse Tartrate-Resistant Acid Phosphatase 5b (TRACP-5b, MyBiosource, #MBS701767) levels according to the manufacturer's protocols.

**Cell culture**. HEK293T and C3H10T1/2 cells were obtained from ATCC (C3H10T1/2, Clone 8, and CCL-226TM). HEK293T cells were maintained in Dulbecco's modified Eagle's medium (DMEM) supplemented with 10% fetal bovine serum. C3H10T1/2 cells were cultured with Eagle's Basal Medium (BME) plus 10% heat inactivated fetal bovine serum. Mesenchymal stem cells (MSCs) isolated from the bone marrow of mice, as reported previously[63], were cultured with α-MEM medium supplemented with 15% heat inactivated embryonic stem cell qualified fetal bovine serum, 0.22 g NaHCO$_3$/100 ml media and 1% Penicillin/Streptomycin/L-Glutamine (P/S/G, Corning Product Number:30-002-CI). Mouse Embryonic Fibroblasts (MEFS) were maintained in Dulbecco's Modified Eagle's Medium (DMEM) plus 10% heat inactivated fetal bovine serum, 1% P/S/G and 1% MEM Nonessential Amino Acids (Corning cellgro, #25-025-CI). MSCs were serum starved with a-MEM medium supplemented with 0.5% heat inactivated embryonic stem cell qualified fetal bovine serum, 0.22 g NaHCO$_3$/100 ml media and 1% P/S/G, whereas C3H10T1/2 cells and MEFs were serum starved with reduced serum medium OPTI-MEM I (1×) (Gibco, 31985070). For osteoblast differentiation,

C3H10T1/2 cells and MSCs were treated with Mesencult ™ Osteogenic Stimulatory Kit (STEMCELL TECHNOLOGIES, #05504). Furthermore, MSCs were treated with Mesencult ™ Adipogenic Differentiation Kit (STEMCELL TECHNOLOGIES, #05507) in order to induce adipogenic differentiation. The isolation of MSCs from the bone marrow of 4-weeks-old mice was performed based on an established protocol for the isolation and culture of these cells[63].

Mouse embryonic fibroblast cultures were prepared as follows: pregnant females were euthanized when embryos were 13.5–14.5 days old. The uterine horns containing embryos were removed from the mouse and placed in a dish contained sterile PBS on ice. Then the dish was transferred to a new dish that contained PBS in the hood. The uterine wall was teared opened and embryos were moved to 6-well dishes contained sterile PBS. The head of the embryo was removed and used for genotyping. The red tissue was removed in the body cavity and discarded. The remainder of the embryo was used to make MEFS. Each embryo was minced with forceps and transferred to a 15 ml conical flasks containing 5 ml of trypsin. Furthermore, the cells got into suspension by pipetting up and down and the flask stayed in 37 °C incubator for 5 min. The cell suspension was transferred in another tube with pre-warmed MEF medium, and the aforementioned process was repeated four times in order to resuspend all the cells. Then the cells were centrifuged and the pellet was resuspended in 25 ml media and plate in a 150 mm plate.

**Reagents**. MLN4924 (Pevonedistat, # HY-70062) was purchased from MedChemExpress (MCE). Paraformaldehyde (#J19943-K2) and Prolonged Diamond DAPI (#P36966) were purchased from Thermofisher. Finally, 4-hydroxytamoxifen was purchased from Sigma (#H6278) and SAG was purchased from Abcam (#ab142160). The working concentration of: (a) SAG is 1 μg/ml, (b) 4-hydroxytamoxifen for the ex vivo experiments is 2 μg/ml, and (c) MLN4924 is 0.25 or 0.5 μM.

**Plasmids**. hNde1 cDNA was obtained from Open Biosystems and was subcloned into a pFLAG-CMV-2 vector[28]. Myc-GLI2, HA-GLI3P1-4A-FLAG, HA-GLI3P1-6A-FLAG, Myc-GLI2 delta N, MYC-FBW7 plasmid were obtained from Addgene. TALPID3-DDK-MYC was obtained from ORIGENE. FBW7α-HA, FBW7β-HA, FBW7γ-HA, FBW7α-HA$^{\Delta F}$, FBW7α-HA$^{R456C}$ were designed by having a 4 × HA tag at the C-terminus. Gli-BS plasmid was a gift from Dr. Brad Yoder (UAB, AL).

**CRISPR-CAS9 knockout**. C3H10T1/2$^{Fbxw7KO}$ and C3H10T1/2$^{Nde1KO}$ clones were generated as follows: C3H10T1/2 cells were transfected with a lentiviral vector encoding Cas9, resistance to puromycin and single guide RNA (sgRNA) sequences specific for Fbxw7 (5′-CACCGATGAAGTCTCGCTGGAACTG-3′) or Nde1 (5′-CACCGACTCCAGCTCCATGCGAAGG-3′). After transfection, the cells underwent serial dilutions and were plated under puromycin selection (1 μg/ml) for 3 weeks. After selection, single colonies were extracted and grown. For the generation of the C3H10T1/2$^{Fbxw7KO;Nde1KO}$ clone, a C3H10T1/2$^{Fbxw7}$KO clone was co-transfected with the aforementioned lentiviral vector encoding Cas9 resistance to puromycin and a single guide RNA (sgRNA) sequence specific for Nde1 (5′-CACCGACTCCAGCTCCATGCGAAGG-3′), and with an empty vector encoding resistance to G418 in order to be able to select C3H10T1/2$^{Fbxw7KO;Nde1KO}$clones. The G418 concertation that was used for the selection of the double KO clones was 1 mg/ml. For the generation of C3H10T1/$^{Ift88KO}$ clones the same methodology was applied: C3H10T1/2 cells were transfected with a lentiviral vector encoding Cas9, resistance to puromycin and single guide RNA (sgRNA) sequences specific for IFT88 (5′-CACCGCAACCCAGCCTATGATACTG-3′) The resulting clones were evaluated for deletion of the genes of interest by DNA sequencing or/ and Western blotting.

**Transient transfection**. siRNAs were transfected into C3H10T1/2 cells using RNAiMAX reagent (Invitrogen) or using Lipofectamine 2000 (Invitrogen) if co-transfected with plasmids according to the manufacturer's instructions. Transfections with plasmids in C3H10T1/2 were done by using Lipofectamine LTX with PLUS™ Reagent (Invitrogen). Transient transfections were done in 293T cells by using calcium phosphate method[28].

**siRNA sequences**. Mouse Fbxw7-specific smart pool siRNA was obtained from Dharmacon (#L-041553-01-0005). Non-targeting siRNA pool was obtained from Dharmacon (#D-001810-10-05). Mouse Talpid3 -specific smart pool siRNA was obtained from Dharmacon (#L-043740-01-0005).

**Immunoblotting**. HEK293T cells, wild type C3H10T1/2 cells, C3H10T1/2$^{Fbxw7}$KO, C3H10T1/2$^{Nde1KO}$ and C3H10T1/2 $^{Fbxw7KO;Nde1KO}$ clones were lysed in 1% Triton X-100, 150 mM NaCl, 10 mM Tris-HCl at pH 7.5, 1 mM EGTA, 1 mM EDTA, 10% sucrose, and a protease inhibitor cocktail (Roche Applied Science), phosphatase inhibitor cocktail (PhosSTOP EASYpack, ROCHE) at 4 °C for 30 min. Cell lysates were separated with SDS-PAGE. Antibodies were used against ALP (Thermofisher, 1:200), NDE1 (Proteintech, 1:1000), β-tubulin (Santa Cruz, 1:1000), TALPID3 (Proteintech 1:1000), HA (Santacruz 1:1000), MYC-Tag (Cell Signaling), FLAG (Sigma 1:1000), FBW7 (Bethyl Laboratories. A301-720A; A301-721A 1:500), NDEL1 (Abcam, 1:1000),

GAPDH (GeneTex, 1:2000), and IFT88 (Proteintech, 1:000). Densitometric quantification was performed with the Licor Image Studio software.

**Immunoprecipitation**. Wild type C3H10T1/2, Fbxw7-null C3H10T1/2, Nde1-null C3H10T1/2, or double mutant cell lysates were incubated with α-FBW7 antibody (Abnova) overnight to immunoprecipitate FBW7. The antigen–antibody complexes were then incubated with Protein G Sepharose beads for 3 h in 4 °C and were analyzed for the presence of FBW7 by Western blot. The same immunoprecipitation conditions were applied for the interactions between GLI2-NDE1 or TALPID3 and FBW7. In the GLI2–NDE1 interactions cell lysates were incubated with the FLAG antibody, and in the TALPID3–FBW7 interaction cell lysates were incubated either with a-TALPID3 or with a-HA.

**Indirect immunofluorescence**. MSCs, Mouse embryonic fibroblasts and C3H10T1/2 cells were grown on glass coverslips and fixed in 4% paraformaldehyde, permeabilized in 0.1% Triton X-100 in PBS, blocked in 3% heat-activated goat serum (or donkey serum for ALP)/0.1% Triton X-100 in PBS (blocking buffer), and incubated overnight with primary antibodies diluted in blocking buffer at 4 °C. Primary antibodies were used against mouse acetylated a-tubulin at 1:1000 (Sigma Aldrich, 1:1000), CD106 (Abcam, 1:200), or goat ALP (Thermofisher, 1:50). Cells were washed three times with PBS and incubated for 2 h at 4 °C with appropriate combinations of AlexaFluor-conjugated secondary antibodies (Invitrogen,1:2000) for 2 h at 4 °C protected from light. Excess of secondary antibodies were removed by four washes in PBS Samples were mounted with Diamond DAPI (Thermofisher) to counterstain the nuclei. z-stacks were obtained with an Olympus FV1000 confocal microscope and processed with ImageJ software for ciliary length measurements. Quantification of the N:C fluorescence ratio corresponding to GLI2ΔN-myc levels was performed by quantifying mean nuclear and cytosolic fluorescence for each field. Background fluorescence was determined and subtracted from nuclear and cytosolic fluorescence[64].

**Quantitative polymerase chain reaction (qPCR)**. RNA was extracted and purified from Wild type mesenchymal stem cells or Fbxw7 null-MSCs at different time points of osteogenic treatment or at 5 days of adipogenic treatment using Trizol reagent (Invitrogen). RNA was reverse-transcribed to cDNA and samples were amplified by qPCR. mRNA levels of the genes of interest were normalized to wild type via the ΔΔCt method. Primers used for qPCR were Fbxw7 Fw: 5′-ACTGGA GAATTTTGGCTGAGGAT-3′, Fbxw7 Rv: 5′- ATGGGCTGTGTATGAAACCT GG-3′, Runx2 Fw: 5′- CCGAAATGCCTCCGCTGTTA-3′, Runx2 Rv: 5′- TGAAA CTCTTGCCTCGTCCG-3′, OSX Fw: 5′-GATGGCGTCCTCTCTGCTTGA-3′, OSX Rv: 5′- CAGGGTTGTTGAGTCCCGCA-3′, ALP Fw: 5′- GCAAGGACATCGCAT ATCAGC-3′, ALP Rv: 5′- TCCAGTTCGTATTCCACATCAGT-3′, OCN Fw: 5′- AG CGGCCCTGAGTCTG-3′, OCN Rv: 5′-CTGGGCTGGGGACTGA-3′, Optn Fw: AG CTTGGCTTATGGACTGAGG, Optn Rv: AGACTCACCGCTCTTCATGTG, GAP DH: Fw: 5′- AAAATGGTGAAGGTCGGTGTG-3′, GAPDH: Rv: 5′- AATGAAGGG GTCGTTGATGG-3′, CEBP1a: Fw: 5′-GGGAACGCAACAACATCGC-3′, CEBP1a: Rv: 5′- GCGGTCATTGTCACTGGTCA-3′, Adiponectin: Fw: 5′- GCAGAGATGG CACTCCTGGA-3′, Adiponectin: Rv: 5′- CCCTTCAGCTCCTGTCATTCC-3′, PP ARγ: Fw: 5′- GTGGGGATAAAGCATCAGGC-3′, PPARγ: Rv: 5′- TCCGGCAG TTAAGATCACACC-3′, TBP: Fw: 5′-TCTACCGTGAATCTTGGCTGT-3′, TBP: Rv: 5′- GTCCGTGGCTCTCTTATTCTCA-3′.

**Osteogenic differentiation**. Osteogenic differentiation induction was performed as follows: Mesenchymal stem cells or C3H10T1/2 cells were trypsinized and seeded in a 96-well or 24-well plate, respectively. When 100% confluency was reached, the medium was replaced with osteogenic induction medium (STEMCELL TECHNOLOGIES, #05504). The osteogenic differentiation for mesenchymal stem cells was evaluated by Alizarin Red S staining at days 14 and 21, and 28 days of differentiation and quantitative real-time PCR of various osteogenic marker genes at these days. Alkaline phosphatase (ALP) staining was used as an indicator of osteoblast differentiation in C3H10T1/2 at 5 days of osteogenic treatment. Triplicate tests were conducted in each experiment.

**Alizarin Red S staining**. Mesenchymal stem cells were differentiated to osteoblasts with the osteogenic induction medium. Then, cells were washed with PBS, fixed with 4% paraformaldehyde and incubate at room temperature for 15 min. The fixative was then removed and cells were washed twice with PBS. Then samples were incubated with Alizarin Red Solution at room temperature for 20 min. After 20 min of incubation, the dye was removed and the cells were gently washed three times with ddH20. After the last wash plates were let to dry and were ready for image acquisition at the dissection microscope.

**Alkaline phosphatase staining**. Undifferentiated C3H10T1/2 cells show weak alkaline phosphatase (ALP) activity whereas differentiated osteoblasts show very high activity of ALP. ALP activity is therefore a good indicator of osteoblast differentiation. One BCIP/NBT tablet (SigmaFast™ BCIP-NBT, Sigma Aldrich) was dissolved in 10 ml distilled ddH20 to prepare the substrate solution, which detects ALP by staining the cells blue-violet when ALP is present. It should be stored in the

dark and used within 2 h. The washing buffer was made by adding 0.05% Tween 20 to PBS without $Ca^{++}/Mg^{++}$ (Corning, #21-040-CM). Cells were washed one time with PBS and fixed with 4% paraformaldehyde for 60 s. After the fixation, cells were washed with washing buffer and then treated with BCIP/NBT substrate solution at room temperature in the dark for 10 min. The staining process is being observed every 2–3 min. Cells are then washed with 1× with washing buffer and 1× with PBS, respectively. After the last wash, the staining results were analyzed at the dissection microscope.

**Adipogenic differentiation**. Adipogenic differentiation induction was performed as follows: Mesenchymal stem cells were trypsinized and seeded in a 96-well plate respectively. When 100% confluency was reached, the medium was replaced with Mesencult TM Adipogenic Differentiation Kit (STEMCELL TECHNOLOGIES, #05507). The adipogenic differentiation for mesenchymal stem cells was evaluated by Oil Red O staining and quantitative real-time PCR of adipogenic marker genes at 5 days of differentiation.

**Oil O Red staining**. Oil Red O was purchased from Sigma (#O-0625). The stock solution was made by adding 0.35 g Oil Red O in 100 ml of isopropanol. The Oil Red O working solution was made by mixing 6 ml of Oil Red O stock solution with 4 ml of $ddH_2O$, let it at room temperature for 20 min followed by filtering. Cells were washed 1× with PBS and fixed with 4% of paraformaldehyde for 15 min at room temperature. Cells were washed twice with $ddH_2O$ and then 1× with 60% isopropanol for 5 min at RT. After the aspiration of isopropanol cells should be dried completely and then incubated with Oil Red O working solution for 10 min at room temperature. The next step is the washing of cells 4× with $ddH_2O$ and the acquisition of the images acquisition of the images at the dissection microscope.

**Hedgehog activity**. Hedgehog activity was analyzed in C3H10T1/2 or HEK293T cells by using a GLIBS reporter plasmid, expressing firefly luciferase in a manner dependent on the activity of Hedgehog pathway. Full growth medium was replaced 24 h after transfection and cells were lysed in lysis buffer (Promega) 48 h after transfection. Cell lysates were incubated with Stop & Glo substrates (Promega) and luciferase activity was measured in a Synergy Neo2 reader (Biotek).

**In vitro ubiquitinylation assay**. HEK293T cells were transiently co-transfected with TALPID3-DDK-MYC, pcB6-His ubiquitin (provided by R. Baer) and FBW7α-HA, FBW7α-HA$^{\Delta F}$ or FBW7α-HA$^{R456C}$. Twenty-four hours after transfection, cells were treated with 10 μM MG132 (Peptides international) for 5 h, cells were lysed and briefly sonicated in Buffer A containing 100 mM $Na_2PO_4$, 10 mM Tris-HCl, 6 M guanidine-HCl, and 10 mM imidazole pH 8.0. His-Tagged ubiquitylated proteins were immunoprecipitated using nickel-NTA resin (Qiagen) for 2–3 h at room temperature (24 °C). Ni-NTA beads were washed three times with buffer A, two times with washes A/TI (1 volume buffer A and 1 volume TI buffer) (TI buffer, 25 mM Tris-Cl pH 6.8 and 20 mM imidazole), and then wash with TI buffer[9]. Finally, Ni-NTA beads were eluted in SDS loading buffer containing 200 mM imidazole, separated by SDS-PAGE, and detected by immunoblotting.

**Statistics and reproducibility**. The statistical analyses were performed by Software GraphPad Prism 8.3. Student's t-test or one-way ANOVA were performed for the quantification analysis of the results based on the number of comparison groups, followed by the proper ad hoc test. Regarding ad hoc tests, when each mean was compared only to a control mean, Dunnett's test was used. When comparisons were performed in additional pairs of means and not only to a control mean, Sidak's test was used.

**Reporting summary**. Further information on research design is available in the Nature Research Reporting Summary linked to this article.

## Data availability
Data source file is available as Supplementary Data 1. Uncropped gels are available as Supplementary data following Supplementary Figures. All other data are available from the corresponding author on reasonable request.

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

## Acknowledgements

We thank Dr. Lorin Olson for comments on the manuscript. This work was supported by AR064211 (L.T. and M.B.H.), DK59599 (L.T.), and DK12165601 (L.T. and W.C.) from NIH; VA-BLR&D I01BX003453 (W.C.) from VA, Presbyterian Health Foundation (M.B.H.), and Oklahoma Center for Adult Stem Cell Research (L.T.).

## Author contributions

E.P. performed in vitro, ex vivo and vivo experiments. V.G. performed GLI2/NDE1 immunoprecipitation experiments. N.S. performed microcomputing analysis on isolated bones. E.P., W.C., M.B.H., and L.T. analyzed data. E.P. and L.T. wrote the manuscript.

## Competing interests

The authors declare no competing interests.
