## [Peer Review File · Communications Biology]

Reviewers' comments:

Reviewer #1 (Remarks to the Author):

Ciliary length alterations are associated with ciliary defects. This work entitled "FBW7 couples structural integrity with functional output of primary cilia" aims to reveal how disorders of ciliary length lead to functional outcomes. By using the Hedgehog pathway and mesenchymal stem cell differentiation to osteoblasts as a read-out to study ciliary structure-function relationships, Petsouki and colleagues identified FBW7 as a regulator of NDE1, a positive regulator of ciliogenesis, and of TALPID3, a negative regulator of ciliogenesis, and showed/suggested different roles of the three proteins in the regulation of Hedgehog signaling.

The topic analysed in this work is of current importance and of interest to a broad readership. The studies provided in this report were carefully performed and are presented in a comprehensible manner. Nevertheless, there are several points which have to be improved before the manuscript is suitable for publication in *Communications Biology*.

1) Line 49: The given references are associated to FBW7 and cilia length. But the sentence is a common statement about ciliary length and signaling. For this reason, additional publications should be indicated, e.g. Cui et al., 2011, Dowdle et al., 2011, Garcia-Gonzalo et al., 2011, Ishikawa and Marshall, 2011 and Larkins et al., 2011 (Cui et al., 2011; Dowdle et al., 2011; Garcia-Gonzalo et al., 2011; Ishikawa and Marshall, 2011; Larkins et al., 2011).

2) Line 55: References are missing for the sentence: "In the absence of the Hedgehog ligand, GLI2 and GLI3 transcription factors are proteolytically processed at the base of the cilium to generate transcriptional repressors, GLI2R and GLI3R. Adequate references would be: Gerhardt et al., 2015, Haycraft et al., 2005 and Wen et al., 2010 (Gerhardt et al., 2015; Haycraft et al., 2005; Wen et al., 2010).

3) The quantifications of cilia length are central investigations in this study. However, the ciliary length measurements were performed by using only one cilia marker (acetylated α -tubulin). Electron microscopy would be helpful to confirm the ciliary length quantifications. Alternatively, the experiments could be performed by using a second cilia marker [e.g. ARL13B (ciliary membrane marker) or IFT88 (component of the intraflagellar transport)] and by performing immunofluorescence microscopy.

In the Figure legends, the number of analyzed cilia is missing in the cilia length quantifications. The number of investigated cells is given but this makes only sense in the ciliation measurements.

4) The authors described an altered cilia length in Fbxw7-deficient C3H10T1/2 cells. Is cilia length also altered in Fbxw7-deficient MEFs? The representative image in Figure 2C does not show any cilium but the quantification of ciliated cells reveals that over 30% of the cells possess a cilium. Accordingly, the representative image should be changed to show that the number of cilia is reduced but that there are still cilia present in some Fbxw7-deficient MEFs.

5) In Figure 4, Petsouki et al. showed cilia length quantifications in different mutant situations in C3H10T1/3 cells. It seems so as if the different mutants have been exclusively compared to the wildtype but not to the other mutants. Or is there no significance between the different mutant situations?

6) GLI2 was overexpressed in Nde1-null C3H10T1/3 cells. Does the transfected full length GLI2 localize to primary cilia?

The authors proposed that "the positive effect of NDE1 on GLI2 activity is cilia-independent". What happens to Hedgehog Signaling in Nde1-deleted cells when IFT88 (or another protein such as KIF3A) is deleted? This would be an important analysis to test this hypothesis.

7) This manuscript is broadly focused on the effect of FBW7 on cilia structure and cilia function. The authors detected cilia length alterations in several mutant cell types. Fbxw7-null C3H10T1/3 cells showed shorter cilia, while Nde1-null C3H10T1/3 cells displayed longer cilia. In both cases, osteoblast differentiation was severely reduced. In Fbxw7/Nde1 double mutants, osteoblast

differentiation was partially restored. By performing experiments with Ciliobrevin A and Ift88-specific sgRNA, Petsouki et al. suggested that the deletion of Fbxw7 and Nde1 led to the formation of functional cilia. Moreover, the authors mentioned that "TALPID3 is required for early stages of cilia formation and organization of transition fibers to assemble the ciliary gate at the base of the cilium (Kobayashi et al., 2014)" and demonstrated that the depletion of Talpid3 in Fbxw7/Nde1 double mutants reduced osteoblast differentiation. In this context, it would significantly improve the quality of the manuscript when the authors analyze transition zone assembly as it was done before by others (Garcia-Gonzalo et al., 2011; Shi et al., 2017; Wiegering et al., 2018). Considering the transition zone assembly hierarchy in vertebrates (Garcia-Gonzalo et al., 2011; Wiegering et al., 2018), Petsouki and colleagues should quantify at least the amount of TCTN2, RPGRIP1L and CEP290 at the ciliary transition of wildtype C3H10T1/3 cells, of wildtype C3H10T1/3 cells with transient depletion of Talpid3, of Fbxw7-null C3H10T1/3 cells, of Nde1-null C3H10T1/3 cells, of double mutant Fbxw7/Nde1-null C3H10T1/3 cells and of double mutant Fbxw7/Nde1-null C3H10T1/3 cells with transient depletion of Talpid3. Although cilia structure alterations are highlighted as one of the most important findings in this study, they have not been analyzed in more detail. By investigating transition zone assembly as suggested, the authors would provide a more comprehensive insight into the relationship between FBW7 and cilia structure.

References:

- Cui, C., Chatterjee, B., Francis, D., Yu, Q., SanAgustin, J., Francis, R., Tansey, T., Henry, C., Wang, B., Lemley, B., et al. (2011). Disruption of Mks1 localization to the mother centriole causes cilia defects and developmental malformations in Meckel-Gruber syndrome. *Dis. Model. Mech.* 4, 43-56.
- Dowdle, W., Robinson, J., Kneist, A., Sirerol-Piquer, M., Frints, S., Corbit, K., Zaghoul, N., van Lijnschoten, G., Mulders, L., Verver, D., et al. (2011). Disruption of a ciliary B9 protein complex causes Meckel syndrome. *Am. J. Hum. Genet.* 89, 94-110.
- Garcia-Gonzalo, F., Corbit, K., Sirerol-Piquer, M., Ramaswami, G., Otto, E., Noriega, T., Seol, A., Robinson, J., Bennett, C., Josifova, D., et al. (2011). A transition zone complex regulates mammalian ciliogenesis and ciliary membrane composition. *Nat. Genet.* 43, 776-784.
- Gerhardt, C., Lier, J., Burmühl, S., Struchtrup, A., Deutschmann, K., Vetter, M., Leu, T., Reeg, S., Grune, T. and Rütger, U. (2015). The transition zone protein Rpgrip1l regulates proteasomal activity at the primary cilium. *J. Cell Biol.* 210, 115-133.
- Haycraft, C., Banizs, B., Aydin-Son, Y., Zhang, Q., Michaud, E. and Yoder, B. (2005). Gli2 and Gli3 localize to cilia and require the intraflagellar transport protein polaris for processing and function. *PLoS Genet.* 1, e53.
- Ishikawa, H. and Marshall, W. (2011). Ciliogenesis: building the cell's antenna. *Nat. Rev. Mol. Cell Biol.* 12, 222-234.
- Kobayashi, T., Kim, S., Lin, Y., Inoue, T. and Dynlacht, B. (2014). The CP110-interacting proteins Talpid3 and Cep290 play overlapping and distinct roles in cilia assembly. *J. Cell Biol.* 204, 215-229.
- Larkins, C., Aviles, G., East, M., Kahn, R. and Casparly, T. (2011). Arl13b regulates ciliogenesis and the dynamic localization of Shh signaling proteins. *Mol. Biol. Cell* 22, 4694-4703.
- Shi, X., Garcia, G. r., Van De Weghe, J., McGorty, R., Pazour, G., Doherty, D., Huang, B. and Reiter, J. (2017). Super-resolution microscopy reveals that disruption of ciliary transition-zone architecture causes Joubert syndrome. *Nat. Cell Biol.* 19, 1178-1188.
- Wen, X., Lai, C., Evangelista, M., Hongo, J., de Sauvage, F. and Scales, S. (2010). Kinetics of hedgehog-dependent full-length Gli3 accumulation in primary cilia and subsequent degradation. *Mol. Cell Biol.* 30, 1910-1922.

Wiegering, A., Dildrop, R., Kalfhues, L., Spsychala, A., Kuschel, S., Lier, J., Zobel, T., Dahmen, S., Leu, T., Struchtrup, A., et al. (2018). Cell type-specific regulation of ciliary transition zone assembly in vertebrates. *EMBO J.* 37, pii: e97791.

Reviewer #2 (Remarks to the Author):

The paper describes deletion of *Fbw7* in MSCs suppresses osteoblast differentiation, resulting in reduced bone formation and mild osteopenia in 3-month-old mice. I can follow this statement well and I feel this conclusion is backed up by the experimental data shown. Authors show reduction in the number of primary cilia in different cell lines from *Fbw7* ko mice as well as using siRNA against *Fbwx7*. siRNA treated cells also seem to have shorter cilia.

However, the following conclusions are rather difficult to grasp and I have difficulties to see the experimental proof of the conclusions made, they seem largely hypothetical and often appear quite twisted.

There is a number of experimental shortfalls (please see my major points for details) and conclusions appear somewhat twisted. Lack of description of some experimental procedures adds to this but is not the major cause. Rather, causality is attributed to findings without direct proof and conclusions are drawn by indirect results.

To give one of many examples, the authors state that while both *NDE1* and *Fbwx7* Null cells exhibit inhibited osteogenic differentiations, authors find this restored in the double knockout. From this they conclude that *FBW7*-mediated degradation of *NDE1* is essential for osteoblast differentiation. No experimental data is provided showing *NDE1* degradation in the presence of *Fbwx7* in the cells used. Even if *Fbwx7* induces *NDE1* degradation in these cells, there is no evidence provided that this process is essential for osteogenic differentiation. Differentiation in the double KO may occur through a totally different mechanism.

The model proposed also in the abstract "which *FBW7* controls the abundance of positive and negative regulators of ciliogenesis impacting on Hedgehog pathway activity and thus, mediating structure-function relationship in primary cilia" cannot be concluded from the data provided.

Major points

Results:

-*Fbw7* ko mice BALP: The authors write that reduced Serum bone-specific alkaline phosphatase levels in the 12 weeks old ko animals compared to controls indicate reduced bone mass may have been primarily caused by reduced osteoblast differentiation and/or function. Alkaline phosphatase is a marker of bone metabolism. It increases with bone turnover, which can be increased osteoclast activity. That lower AP is a result of reduced osteoblast differentiation is not supported experimental by data, it may also result from reduced osteoclast activity.

-Reduced osteogenic differentiation in *Fbw7* ko MSCs:

-I agree that Alizarin red staining in Fig2b KO suggests reduced differentiation in KO cells. However already at day 14, wt cells seem at maximum alizarin red intensity, there is no image shown before the onset of differentiation induction as a control. To judge cell density is equal at the start, providing DIC images of live cells eg in supp would also help.

-Figure 3a: There is rather a lot of non-ciliary tubulin stained, also in the wt image (eg several cilia-like red structures on one nucleus). I would suggest to use *Arl13b* as an axonemal marker to avoid staining of cytosolic tubulin. Currently its not clear what is a cilium and what is cytosolic tubulin except for the image of *Nde1* ko.

-Figure 3b: see comments on 3a, its not clear what is a cilium and what is cytosolic acet tubulin, therefore I am not sure the statistics reflects cilia formation abilities of the cell lines.

-Figure 3c: To judge this is a general effect on cilia length, I would suggest to provide images with more than one cell on it.

-*Nde1* ko: authors write does not affect cilia number but increases ciliary length as the previously

published, however no any actual data from the current experiment is provided. This is essential.

-FBW7 regulates functional integrity of cilia in C3H10T1/2 cells: The authors describe use of Ift88 ko cells lacking cilia to investigate the necessity of cilia for osteoblast differentiation. As a marker they use AP. I would suggest to use the same qPCR markers as used in the Fbw7 ko cells (Runx2 etc) if patterns are compared between the lines and if a conclusion is made that lack of differentiation in FBWx7 cells is a result of cilia dysfunction.

-C3H10T1/2 cells NDE1 LOF cells: Its not clear from the paper or the sup if this is a mixed CRISPR edited cell line used for in vitro experiments or single clones? If it is a mixture, no data on targeting efficiency is provided. If it's a single clone versus a single control clone, more clones need to be tested (minimum 3 of each, better 5) to define if the difference observed is a result of NDE1 LOF or a clonal effect. Genotypes of the clones need to be provided.

-Figure 5: Authors write that to test if cilia were functional in Fbw7/NDE1 double Ko cells, they destroyed cilia either pharmacologically or by ift88 targeting. No data of ift88 targeting efficiency and effect on ift88 protein levels is provided however. Authors conclude from a lower AP level after "anti-cilia" treatment compared to before treatment that cilia were functional before. There is no direct evidence that AP can be used as marker of ciliary function.

-the reduced level of differentiation in Nde1-null cells was unexpected based on effects on cilia alone and suggested that knocking out NDE1 uncoupled structure from function."': That's hard to understand, there are longer cilia in the KO which is a structural defect why is function uncoupled from structure? Maybe what this is meant to say is that also a normal looking cilium can be dysfunctional. This is not new, there is plenty of examples eg for fibroblasts of human ciliopathy patients with normal looking cilia and for example impaired hedgehog signalling. However, NDE1 cilia are described as structurally ABNORMAL by the authors and it currently also implies that an abnormal ciliary structure can be fully functional for which to my knowledge there is no proof.

-Figure6A: Authors write that cell lines were transfected with Gli plasmids. No data on equal expression levels is provided. Graph reads Hedgehog pathway activity, the legend says its luciferase activity, was this and if yes to what was it normalised? This data is essential.

-I would recommend to use another method of hedgehog signalling activity assessment eg western blot for gli3 repressor and activator as well as IF for Smo and Gli in addition.

-"Full length or constitutively active GLI2 increased differentiation in these cells, suggesting the possible defects in GLI2-mediated signaling may account for the impaired cilia functionality in Nde1-null cells. Interestingly, overexpression of a constitutively active fulllength GLI3 construct (GLI3 P1-P6) had no effect in Nde1-null cells, while acting as a dominant negative allele in Fbw7-null cells (Supplementary Fig. 5)"

These conclusions are hard to follow. Hedgehog signalling requires a functional cilium as stated by the authors in the introduction. Hedgehog effects depend on the balance between repressor and active forms, achieved by cleavage. While overexpression of Gli can increase overall activity, it seems unlikely this happens in a dysfunctional cilium. The fact Gli overexpression rescues NDE1 hedgehog effect may point towards a non-ciliary origin of the problem in these cells.

The term "dominant negative" effect is usually used if a dysfunctional gene product inhibits the function of a also present functional product. Here, Fbw7 is dysfunctional, not Gli3. What the authors mean I guess is that overexpression of Gli3 did not enhance hedgehog signalling but repressed it. This needs to be rephrased.

Why did Gli3 overexpression have a different effect from Gli2 overexpression? Is there a hypothesis?

Figure 6f: co-IP NDE1 and Gli2: There is no adequate negative control, please use a Flagtagged protein with Gli2-myc as control

Luciferase assays: no expression control provided/no normalization?

-“These data could help explain the reduced overall GLI activity in Nde1-null cells and further indicate that while NDE1 functions as a negative regulator of ciliogenesis, it functions as a positive regulator of cilia based signaling such as Hedgehog signaling, coupling ciliary structure and function.”

What is written here suggests the opposite, it would uncouple function (hedgehog) and structure.

“TALPID3 is required for early stages of cilia formation and organization of transition fibers to assemble the ciliary gate at the base of the cilium, which would be consistent with the small but significant increase in ciliary length in double mutant cells.”

Ciliogenesis is not a process where a lot of proteins each contribute a nm of axonemal length. Ciliary proteins each have different essential roles. I cannot follow this hypothesis.

Figure 7 co-IPs: Again no adequate negative controls. h-k: again ALP used as differentiation marker instead of the markers used for Fbw7 KO. Only one marker is insufficient, Fbxw7 markers should be used.

-“These data suggest FBW7 controls the abundance of TALPID3”: there is no influence on the abundance demonstrated nor accumulation of TALPID3 in double knockout cells as suggested by the abstract.

-Minor points:

Introduction:

-I would suggest to replace "Maximal hedgehog activity" by "adequate hedgehog activity". Mouse models eg Ift144 suggest that cilia dysfunction can also result in a broadened hedgehog activity pattern, so cilia do not necessarily increase hedgehog activity overall, they are more responsible for properly controlled adequate hedgehog activity

Results:

-Its confusing that Title and large parts of the introduction use FBW7 as a term and at the beginning of the results, Fbw7 is used. I would suggest to use a singular term or add a very short explanation in the introduction.

-“However, since Nde1-null cells have normal ciliation and 202 even longer primary cilia compared to wild type, we hypothesized that functionality of cilia203 related Hedgehog effectors might have been compromised.” Again, longer than normal cilia suggest a structural problem which could in turn cause a functional problem.

Figure 7A co-IP: Why are both proteins my tagged in this co-IP, authors use an HA version in the figure part on the right as well?

Reviewer #3 (Remarks to the Author):

The authors seek to identify the role of two components previously identified as regulators of primary cilium formation/function in mesenchymal stem cell differentiation and, ultimately, bone tissue maintenance. In this work, they reveal that a protein network consisting of Fbxw7/Nde1/TALPID3 coordinates primary cilium function and Hh signaling (via the cilium or independently) to encourage osteogenic differentiation. Overall, the manuscript is very well written and organized, the experimental design is clear and appropriate, and the authors' conclusions are supported by their data. Furthermore, the results are novel for the primary cilium and bone cell fields and will likely contribute to further investigation by other researchers. My comments to strengthen the scientific rigor and novelty of the manuscript are outlined below.

Abstract: My only comment is regarding precise language. The authors do a great job of using the full term "primary cilia" at the beginning of most paragraphs in the rest of the manuscript, but the general term "cilia" is exclusively used in the abstract. It is important to distinguish between primary and motile cilia since these structures and research fields can be very different. I recommend clarifying primary cilia in the beginning of the abstract at the very least and

thoroughly checking the rest of the document for precise use of these terms.

Introduction: Well organized and clearly written. All the necessary terms are defined and an appropriate amount of background information is provided to set up the reader.

Methods: In general, the methods provide enough detail that experiments can be recapitulated but two key descriptions are missing. First, it is insufficient to simply state "followed by the proper ad hoc test" in the statistics section. According to the figure captions, I only see Dunnett's being used so please explicitly state this in the methods and provide a brief rationale for selecting this ad hoc test. Second, there is no description of how cilium length was measured other than to say it was done using ImageJ. This is not a trivial process and the measurements can be drastically influenced by how length is defined (i.e. did the authors measure in 2D or generate 3D z-stacks to visualize cilia length?), so it is critical for the authors to include this information in their methods.

Results:

Figure 1: What is the justification for only using male mice? It is well known that sex can influence bone phenotype so it is critical for the authors to provide their reasoning for using only one sex. Moreover, if there is no phenotype in female mice this is important information to disclose. Was cortical bone examined in these mice? C57BL6/J mice famously have very dense cortical bone and are known to not display cortical phenotypes, but many bone researchers would be curious to know if a difference exists.

Figure 2: Consider using the term "cilium incidence and length" rather than "ciliation". This is more precise wording for % of ciliated cells as a whole -- rather than whether an individual cell has a primary cilium -- and more accurately describes your experimental read outs.

Figure 3: This experiment is well designed, I just have a suggestion regarding data presentation. The multiple bar graphs in (c),(d), and (e) could be neatly organized onto a single plot with the columns for each target gene spaced apart and labeled since they all share the same scale and y-axis label. Figure 3 from a recent publication is a good example of what I am trying to explain: <https://www.nature.com/articles/s42003-021-01675-4>. Currently, the bar graphs fill an unnecessary amount of space and the redundant labels/ axes visually take away from the otherwise great data.

Figure 4: Great, no comments.

Figure 5: I like the concept of this experiment but I have several issues with execution and presentation. First, it is already well established that cells require primary cilia to undergo osteogenic differentiation. In lines 169-170, the wording indicates this is a new finding, when really this experiment simply demonstrates that their techniques successfully disrupted primary cilia. Furthermore, I am confused as to why Ciliobrevin A treatment was introduced. If the authors want to attribute findings to the primary cilium, Ift88 deletion is the most accepted way to uniquely disrupt the cilium. Ciliobrevin is known to influence non-ciliary pathways (<https://www.nature.com/articles/nature10936>) and can confound interpretation of the data. The Ift88 sgRNA data alone are sufficient to demonstrate the authors' point so I would consider moving all Ciliobrevin A data to a Supplementary Figure unless the authors plan to comment on differences between the two approaches. Second, Figure 5a/c and b/d are redundant: measuring ALP using a fluorescent stain does not tell the reader anything more than measuring it with a colorimetric stain did. Furthermore, in 5c the intensity of the WB band and the intensity of the ALP staining for the double KO do not appear consistent to my eye. I would expect the band to be much darker since the ALP stain coloration is comparable to the control, such as is the case in Figure 6b/c. I would recommend editing out the unnecessary data in this figure (i.e. the Ciliobrevin A treatment and select either a/c or b/d) and adding ICC looking at primary cilia and ALP simultaneously. This image would reveal whether ALP production by a single cell correlates with the existence of and/or length of the primary cilium. This will help demonstrate whether or not Nde1 and Fbxw7 directly influence ALP production via the cilium.

Figures 6 and 7 are experimentally sound and provide insights into the individual deletions of Nde1 and Fbxw7, but I am left skeptical that TALPID3 explains the partial recovery in the double KO. I

think this can be resolved by presenting the bar graphs in e, f, h, i, k, and l differently. Rather than showing the reduction in Hh signaling and ALP activity normalized to either WT or the double KO cells, everything should be normalized to the WT cells. This way we can see if the double KO cell readouts are slightly decreased compared to WT and recapitulate the data (especially the ALP readouts) presented in earlier figures. The way the bar graphs are currently presented, all it shows is that knockdown of TAPID3 reduces Hh signaling and ALP in both cell types. It would further strengthen the authors' claims to increase TAPID3 expression and observe whether ALP and Hh signaling are enhanced in WT and double KO cells.

Figure 8: The cartoon is logical based on the authors' conclusions.

Discussion: The authors succinctly summarize their results, provide reasonable speculations based on their data (i.e. enhancement of TAPID3-mediated Hh signaling via deletion of Fbxw7 can overcome the absence of Nde1-mediated Hh signaling), and nicely discuss how their data informs the existing literature and research fields.

We thank the reviewers for constructive comments and helpful suggestions. In the revised version, we addressed the reviewers' comments. Responses are shown in bold.

Reviewer #1:

Ciliary length alterations are associated with ciliary defects. This work entitled "FBW7 couples structural integrity with functional output of primary cilia" aims to reveal how disorders of ciliary length lead to functional outcomes. By using the Hedgehog pathway and mesenchymal stem cell differentiation to osteoblasts as a read-out to study ciliary structure-function relationships, Petsouki and colleagues identified FBW7 as a regulator of NDE1, a positive regulator of ciliogenesis, and of TALPID3, a negative regulator of ciliogenesis, and showed/suggested different roles of the three proteins in the regulation of Hedgehog signaling.

The topic analysed in this work is of current importance and of interest to a broad readership. The studies provided in this report were carefully performed and are presented in a comprehensible manner. Nevertheless, there are several points which have to be improved before the manuscript is suitable for publication in Communications Biology.

1) Line 49: The given references are associated to FBW7 and cilia length. But the sentence is a common statement about ciliary length and signaling. For this reason, additional publications should be indicated, e.g. Cui et al., 2011, Dowdle et al., 2011, Garcia-Gonzalo et al., 2011, Ishikawa and Marshall, 2011 and Larkins et al., 2011 (Cui et al., 2011; Dowdle et al., 2011; Garcia-Gonzalo et al., 2011; Ishikawa and Marshall, 2011; Larkins et al., 2011).

Suggested references were added.

2) Line 55: References are missing for the sentence: "In the absence of the Hedgehog ligand, GLI2 and GLI3 transcription factors are proteolytically processed at the base of the cilium to generate transcriptional repressors, GLI2R and GLI3R. Adequate references would be: Gerhardt et al., 2015, Haycraft et al., 2005 and Wen et al., 2010 (Gerhardt et al., 2015; Haycraft et al., 2005; Wen et al., 2010).

Suggested references were added.

3) The quantifications of cilia length are central investigations in this study. However, the ciliary length measurements were performed by using only one cilia marker (acetylated α -tubulin). Electron microscopy would be helpful to confirm the ciliary length quantifications. Alternatively, the experiments could be performed by using a second cilia marker [e.g. ARL13B (ciliary membrane marker) or IFT88 (component of the intraflagellar transport)] and by performing immunofluorescence microscopy.

In the Figure legends, the number of analyzed cilia is missing in the cilia length quantifications. The number of investigated cells is given but this makes only sense in the ciliation measurements.

We have repeated ciliation and ciliary length measurements using Arl13b as a ciliary marker (please see new Fig. 4g-l). In Figure 4f and 4l, the number of cilia analyzed in length quantifications is shown at the base of each bar in the graph, while the number of cells analyzed for percentage of ciliation is indicated in the legend.

4) The authors described an altered cilia length in *Fbxw7*-deficient C3H10T1/2 cells. Is cilia length also altered in *Fbxw7*-deficient MEFs? The representative image in Figure 2C does not show any cilium but the quantification of ciliated cells reveals that over 30% of the cells possess a cilium. Accordingly, the representative image should be changed to show that the number of cilia is reduced but that there are still cilia present in some *Fbxw7*-deficient MEFs.

We measured ciliary length in wild type and *Fbxw7*-deficient MEFs (please see new Fig. 2c-e). Original, representative images in Fig. 2c have been replaced by new images, showing more cells.

5) In Figure 4, Petsouki et al. showed cilia length quantifications in different mutant situations in C3H10T1/3 cells. It seems so as if the different mutants have been exclusively compared to the wildtype but not to the other mutants. Or is there no significance between the different mutant situations?

We have updated all graphs in Fig. 4 to include the suggested comparisons.

6a) GLI2 was overexpressed in *Nde1*-null C3H10T1/3 cells. Does the transfected full length GLI2 localize to primary cilia?

In our hands, transfected full length wild type GLI2 was undetectable at the primary cilia of wild type and *Nde1KO* cells. However, we transfected wild type and *Nde1KO* cells with a construct expressing an N-terminally truncated form of GLI2 (GLI2 Δ N) that is 30-fold more active than wild type GLI2 in C3H10T1/2 cells¹. We then analyzed the nuclear:cytosolic (N:C) ratio of GLI2 Δ N, following a method reported by Kelley & Paschal². Our results indicated that the fluorescence ratio corresponding to nuclear versus cytoplasmic GLI2 Δ N (N:C ratio) is dramatically reduced in *Nde1KO* cells (please see new Fig. 6g,h). These findings suggest that loss of *Nde1* compromised trafficking of GLI2 to the nucleus, most likely accounting for lower activity of the Hedgehog pathway and delayed osteoblast differentiation. Because this construct encodes an activated form of GLI2, we suggest that NDE1 promotes the nuclear localization of GLI2 at a post-activation stage.

6b) The authors proposed that “the positive effect of NDE1 on GLI2 activity is cilia-independent”. What happens to Hedgehog Signaling in *Nde1*-deleted cells when IFT88 (or another protein such as KIF3A) is deleted? This would be an important analysis to test this hypothesis.

As we have shown previously^{3, 4}, NDE1 overexpression has a robust negative effect on cilia formation, resulting in stumpy or completely absent cilia. We confirmed these results in C3H10T1/2 cells (please see new Suppl. Fig. 9e,f), where overexpression of NDE1 severely suppressed cilia formation in C3H10T1/2 cells. Furthermore, NDE1 overexpression enhanced GLI2 activity in C3H10T1/2 or 293T cells (Fig. 6d,e) without culturing these cells under conditions that would promote cilia formation (ie, serum starvation). Based on these data, we conclude that the positive effect of NDE1 on Hedgehog is cilia-independent or occurs at a post-activation stage of GLI2, which is expected to occur at the cilium. This is consistent with data shown in Fig. 6g,h, where NDE1 affects the nuclear accumulation of GLI2 Δ N.

7) This manuscript is broadly focused on the effect of FBW7 on cilia structure and cilia function. The authors detected cilia length alterations in several mutant cell types. Fbxw7-null C3H10T1/3 cells showed shorter cilia, while Nde1-null C3H10T1/3 cells displayed longer cilia. In both cases, osteoblast differentiation was severely reduced. In Fbxw7/Nde1 double mutants, osteoblast differentiation was partially restored. By performing experiments with Ciliobrevin A and Ift88-specific sgRNA, Petsouki et al. suggested that the deletion of Fbxw7 and Nde1 led to the formation of functional cilia. Moreover, the authors mentioned that “TALPID3 is required for early stages of cilia formation and organization of transition fibers to assemble the ciliary gate at the base of the cilium (Kobayashi et al., 2014)” and demonstrated that the depletion of Talpid3 in Fbxw7/Nde1 double mutants reduced osteoblast differentiation. In this context, it would significantly improve the quality of the manuscript when the authors analyze transition zone assembly as it was done before by others (Garcia-Gonzalo et al., 2011; Shi et al., 2017; Wiegeling et al., 2018). Considering the transition zone assembly hierarchy in vertebrates (Garcia-Gonzalo et al., 2011; Wiegeling et al., 2018), Petsouki and colleagues should quantify at least the amount of TCTN2, RPGRIP1L and CEP290 at the ciliary transition of wildtype C3H10T1/3 cells, of wildtype C3H10T1/3 cells with transient depletion of Talpid3, of Fbxw7-null C3H10T1/3 cells, of Nde1-null C3H10T1/3 cells, of double mutant Fbxw7/Nde1-null C3H10T1/3 cells and of double mutant Fbxw7/Nde1-null C3H10T1/3 cells with transient depletion of Talpid3. Although cilia structure alterations are highlighted as one of the most important findings in this study, they have not been analyzed in more detail. By investigating transition zone assembly as suggested, the authors would provide a more comprehensive insight into the relationship between FBW7 and cilia structure.

We performed the suggested experiments, but we found no obvious differences in the levels of Cep290 and Tctn2 at the TZ of wild type (WT), Fbxw7KO, Nde1KO or double Fbxw7/Nde1KO cells. We could not detect Rpgrip1L.

Figure 1. Expression of Cep290 and Tctn2 in WT, Fbxw7-null (Fbxw7KO), Nde1-null (Nde1KO), and Fbxw7-/Nde1-null (Fbxw7nde1KO) C3H10T1/2 cells.

Depletion of Talpid3 has been shown to have a robust negative effect on ciliation⁵. In our system of C3H10T1/2 wild type and double KO cells, transient depletion of Talpid3 resulted in loss of cilia and elimination of Cep290- or Tctn2-specific signals.

Figure 2. Expression of Cep290 and Tctn2 in wild type or Fbxw7-/Nde1-null (Fbxw7Nde1KO) C3H10T1/2 cells transiently co-transfected with GFP and scrambled (Ctrl) or a mouse Talpid3-specific siRNA.

Reviewer #2:

The paper describes deletion of Fbw7 in MSCs suppresses osteoblast differentiation, resulting in reduced bone formation and mild osteopenia in 3-month-old mice. I can follow this statement well and I feel this conclusion is backed up by the experimental data shown. Authors show reduction in the number of primary cilia in different cell lines from Fbw7 ko mice as well as using siRNA against Fbw7. SiRNA treated cells also seem to have shorter cilia.

However, the following conclusions are rather difficult to grasp and I have difficulties to see the experimental proof of the conclusions made, they seem largely hypothetical and often appear quite twisted. There is a number of experimental shortfalls (please see my major points for details) and conclusions appear somewhat twisted. Lack of description of some experimental procedures adds to this but is not the major cause. Rather, causality is attributed to findings without direct proof and conclusions are drawn by indirect results.

To give one of many examples, the authors state that while both NDE1 and Fbw7 Null cells exhibit inhibited osteogenic differentiations, authors find this restored in the double knockout. From this they conclude that FBW7-mediated degradation of NDE1 is essential for osteoblast differentiation. No experimental data is provided showing NDE1 degradation in the presence of Fbw7 in the cells used. Even if Fbw7 induces NDE1 degradation in these cells, there is no evidence provided that this process is essential for osteogenic differentiation. Differentiation in the double KO may occur through a totally different mechanism.

Please see Supplementary Fig. 5b showing elevated NDE1 levels in Fbxw7KO C3H10T1/2 cells. Our data show that deletion of Fbxw7 suppresses osteoblast differentiation, which is partially restored in double mutant cells. Restored cilia in double mutant cells are functional, in terms of osteoblastic differentiation, as the additional deletion of *Ift88* in double mutant cells causes a suppression of ciliogenesis (Fig. 5e-g). These data led us to conclude that FBW7 has an essential role in osteoblast differentiation, which involves the degradation of NDE1. If FBW7 and NDE1 worked independently, there would be no

rescue in double mutant cells, since in both single mutants, osteoblastogenesis is suppressed.

The model proposed also in the abstract “which FBW7 controls the abundance of positive and negative regulators of ciliogenesis impacting on Hedgehog pathway activity and thus, mediating structure-function relationship in primary cilia” cannot be concluded from the data provided.

NDE1 and TALPID3 are negative and positive regulators of ciliogenesis, respectively and their levels are controlled by FBW7, as shown in this and our previous study⁴. Effects on Hedgehog pathway activity and osteoblast differentiation are also shown in this study (Fig. 6a). Based on these data, it is reasonable to conclude that FBW7 mediates structure - function relationships of primary cilia and its effects on osteoblast differentiation are mediated, at least partially, via cilia. Since, osteoblast differentiation levels are not completely restored in double mutant cells, it is also reasonable to suggest that extraciliary pathways regulated by FBW7 can contribute to its effects on osteoblast differentiation. We acknowledged this possibility in the “Discussion” section.

Major points

Results:

1) Fbw7 ko mice BALP: The authors write that reduced Serum bone-specific alkaline phosphatase levels in the 12 weeks old ko animals compared to controls indicate reduced bone mass may have been primarily caused by reduced osteoblast differentiation and/or function. Alkaline phosphatase is a marker of bone metabolism. It increases with bone turnover, which can be increased osteoclast activity. That lower AP is a result of reduced osteoblast differentiation is not supported experimental by data, it may also result from reduced osteoclast activity.

We agree with the reviewer. However, TRAcP, which reflects osteoclast activity did not change (Fig. 1i). Therefore, the reduction in BALP activity should primarily, if not entirely, reflect reduced osteoblast activity.

2) Reduced osteogenic differentiation in Fbw7 ko MSCs:

- I agree that Alizarin red staining in Fig2b KO suggests reduced differentiation in KO cells. However already at day 14, wt cells seem at maximum alizarin red intensity, there is no image shown before the onset of differentiation induction as a control. To judge cell density is equal at the start, providing DIC images of live cells eg in supp would also help.

Please see new Suppl. Fig. 3a.

3) Figure 4a: There is rather a lot of non-ciliary tubulin stained, also in the wt image (eg several cilia-like red structures on one nucleus). I would suggest to use Arl13b as an axonemal marker to avoid staining of cytosolic tubulin. Currently its not clear what is a cilium and what is cytosolic tubulin except for the image of Nde1 ko.

- Figure 4b: see comments on 4a, its not clear what is a cilium and what is cytosolic acet tubulin, therefore I am not sure the statistics reflects cilia formation abilities of the cell lines.

We performed additional experiments using Arl13b as a ciliary marker. Please also see response to reviewer 1, point 3.

- Figure 4c: To judge this is a general effect on cilia length, I would suggest to provide images

with more than one cell on it.

Images with more cells have been provided in new Fig. 4.

- Nde1 ko: authors write does not affect cilia number but increases ciliary length as the previously published, however no any actual data from the current experiment is provided. This is essential.

Please see new Fig. 4f and 4l. There is no significant difference between wild type, *Nde1KO* and doubleKO C3H10T1/2 cells in terms of ciliation, after 24 h of serum starvation (please see new Fig 4e,k). However, there is a statistically significant increase in ciliary length in the *Nde1KO* and *doubleKO* cells, compared to wild type (Fig. 4e, k). These results also agree with our previously published studies^{3,4}.

4) FBW7 regulates functional integrity of cilia in C3H10T1/2 cells: The authors describe use of Ift88 ko cells lacking cilia to investigate the necessity of cilia for osteoblast differentiation. As a marker they use ALP. I would suggest to use the same qPCR markers as used in the Fbw7 ko cells (Runx2 etc) if patterns are compared between the lines and if a conclusion is made that lack of differentiation in FBWx7 cells is a result of cilia dysfunction.

Please see new Fig. 5d,e. We identified significantly decreased levels of *Osterix* and *Runx2* mRNA levels in *Fbw7KO* and *Nde1KO* clones, whereas levels of these differentiation markers were partially restored in the doubleKO clone.

5) C3H10T1/2 cells NDE1 LOF cells: Its not clear from the paper or the sup if this is a mixed CRISPR edited cell line used for in vitro experiments or single clones? If it is a mixture, no data on targeting efficiency is provided. If it's a single clone versus a single control clone, more clones need to be tested (minimum 3 of each, better 5) to define if the difference observed is a result of NDE1 LOF or a clonal effect. Genotypes of the clones need to be provided.

***Nde1KO* clones used for *in vitro* experiments were single clones. Multiple clones of *Nde1KO* were tested and showed similar levels (reduced) of osteoblast differentiation. Please see new Supp. Fig. 8 for ALP staining and respective genotypes.**

6) Figure 5: Authors write that to test if cilia were functional in Fbw7/NDE1 double Ko cells, they destroyed cilia either pharmacologically or by ift88 targeting. No data of ift88 targeting efficiency and effect on ift88 protein levels is provided however. Authors conclude from a lower ALP level after "anti-cilia" treatment compared to before treatment that cilia were functional before. There is no direct evidence that ALP can be used as marker of ciliary function.

Please see new Fig. 5g.

7) "the reduced level of differentiation in Nde1-null cells was unexpected based on effects on cilia alone and suggested that knocking out NDE1 uncoupled structure from function.": That's hard to understand, there are longer cilia in the KO which is a structural defect why is function uncoupled from structure? Maybe what this is meant to say is that also a normal looking cilium can be dysfunctional. This is not new, there is plenty of examples eg for fibroblasts of human ciliopathy patients with normal looking cilia and for example impaired hedgehog signalling. However, NDE1 cilia are described as structurally ABNORMAL by the

authors and it currently also implies that an abnormal ciliary structure can be fully functional for which to my knowledge there is no proof.

In general, we agree with the reviewer. However, underlying specific mechanisms are not known. Because we saw an inhibition in osteoblast differentiation in *Fbxw7*- cells (reduced ciliation) and *Ift88*-null (no cilia) cells, we did not expect to see the same phenotype in *Nde1*-null cells that showed abnormally long cilia, if the sole function of the FBW7/NDE1 pathway would be to control ciliary elongation. As our new data suggest (please see Fig. 6g,h), deletion of NDE1 compromised the nuclear localization of GLI2 Δ N, which we do not believe is due to abnormally long cilia *per se*, but rather to a specific effect of NDE1 on GLI2 at a post-activation stage. This is supported by: 1) the positive effect of transfected NDE1 on GLI2 in C3H10T1/2 and 293T cells, which were not cultured under conditions that favor efficient cilia formation, 2) strong suppression of cilia formation by transfected NDE1 (Suppl. Fig. 9e), and 3) the physical interaction of NDE1 and GLI2 in transfected 293T cells (Fig. 6f). Therefore, although we do see a similar phenotype in conditions of short or no cilia (ie, *Fbxw7*- or *Ift88*-null cells) and abnormally long cilia (*Nde1*-null cells), underlying mechanisms are different and in the case of abnormally long cilia may not be due to a ciliary defect, *per se*.

8) Figure 6A: Authors write that cell lines were transfected with Gli plasmids. No data on equal expression levels is provided. Graph reads Hedgehog pathway activity, the legend says its luciferase activity, was this and if yes to what was it normalised? This data is essential.

Please see new Supplementary Fig. 9 showing equal expression of GLI2 or GLI3 plasmids. All graphs on Hedgehog activity are provided as normalized to wild type.

9) I would recommend to use another method of hedgehog signalling activity assessment eg western blot for gli3 repressor and activator as well as IF for Smo and Gli in addition.

Previous studies have shown that a downstream effector of the Hedgehog pathway in C3H10T1/2 cells is the canonical Wnt pathway⁶. They further showed that Hedgehog activity induces the expression of *Wnt9a* mRNA. These data are consistent with the roles of these pathways in the determination of segment polarity in flies, where Wnt is downstream of Hedgehog, promoting the production of Wg. Therefore, we used Wnt pathway activity as a readout of Hedgehog activity in wild type, *Fbxw7KO*, *Nde1KO* and *doubleKO* clones. We identified significantly decreased Wnt activity in *Fbxw7KO* and *Nde1KO* clones, whereas Wnt pathway activity was restored in *doubleKO* clones (please see new Fig. 6b). Moreover, we observed a similar pattern in the mRNA levels of *Wnt9a* (also known as WNT14), which has been previously reported to be induced by Hedgehog⁶, to be present in osteoblasts⁷ and to have a positive role in canonical Wnt signaling⁸⁻¹⁰ (please see new Fig. 6c). Therefore, we believe that cilia signaling involves Hedgehog signaling that in turn promotes canonical Wnt/ β -catenin signaling. The positive effect of canonical Wnt/ β -catenin signaling in promoting osteoblastogenesis and suppressing adipogenesis is well-established.

10) "Full length or constitutively active GLI2 increased differentiation in these cells, suggesting the possible defects in GLI2-mediated signaling may account for the impaired cilia functionality in *Nde1*-null cells. Interestingly, overexpression of a constitutively active full length GLI3 construct (GLI3 P1-P6) had no effect in *Nde1*-null cells, while acting as a dominant negative allele in *Fbxw7*-null cells (Supplementary Fig. 5)"
These conclusions are hard to follow. Hedgehog signalling requires a functional cilium as

stated by the authors in the introduction. Hedgehog effects depend on the balance between repressor and active forms, achieved by cleavage. While overexpression of Gli can increase overall activity, it seems unlikely this happens in a dysfunctional cilium. The fact Gli overexpression rescues NDE1 hedgehog effect may point towards a non-ciliary origin of the problem in these cells.

As now shown in new Fig. 6g,h, NDE1 promotes the nuclear localization of GLI2ΔN via a cilia-independent step or a step downstream of its activation, that is expected to occur at the cilium. As Hedgehog pathway activity depends not only on the balance between activators and repressors, but also in trafficking of GLI proteins in the cilium and to the nucleus, deletion of *Nde1* could cause some form of GLI2 trafficking defect, rendering cilia non-functional in terms of proper trafficking of GLI activators, perhaps from the cilium to the nucleus.

The term “dominant negative” effect is usually used if a dysfunctional gene product inhibits the function of an also present functional product. Here, Fbw7 is dysfunctional, not Gli3. What the authors mean I guess is that overexpression of Gli3 did not enhance hedgehog signalling but repressed it. This needs to be rephrased.

We made the correction.

Why did Gli3 overexpression have a different effect from Gli2 overexpression? Is there a hypothesis?

We were intrigued by this finding as well, as it represents a new finding. It suggests that GLI2 is specifically associated with osteoblast differentiation, in these cells. It has been shown that TAF9 interacts with GLI1 and GLI2, but not GLI3 in rhabdomyosarcoma and osteosarcoma cell lines¹¹, which could be relevant here.

11) Figure 6f: co-IP NDE1 and Gli2: There is no adequate negative control, please use a Flag-tagged protein with Gli2-myc as control.

Please see new Supp. Fig. 10 for negative controls.

Luciferase assays: no expression control provided/no normalization?

Results on new Fig. 6d,e are provided as normalized to wild type. We also provide confirmation of expression of the various constructs in these experiments at the bottom of the graphs.

- “These data could help explain the reduced overall GLI activity in *Nde1*-null cells and further indicate that while NDE1 functions as a negative regulator of ciliogenesis, it functions as a positive regulator of cilia based signaling such as Hedgehog signaling, coupling ciliary structure and function.”

What is written here suggests the opposite, it would uncouple function (hedgehog) and structure.

Text was rephrased.

“TALPID3 is required for early stages of cilia formation and organization of transition fibers to assemble the ciliary gate at the base of the cilium, which would be consistent with the small

but significant increase in ciliary length in double mutant cells.”

Ciliogenesis is not a process where a lot of proteins each contribute a nm of axonemal length. Ciliary proteins each have different essential roles. I cannot follow this hypothesis.

The statement is referring to double mutant cells, which show a small but significant increase in ciliary length compared to *Nde1*- single mutants (Fig. 4f,l). Therefore, accumulation of TALPID3 in double mutant cells compared to *Nde1*- single mutants could explain the small additional increase in ciliary length, because of its positive effect on ciliogenesis. We'll be happy to remove this statement, if it is still confusing.

Figure 7 co-IPs: Again no adequate negative controls.

Please see Supplementary Fig. 10.

h-k: again ALP used as differentiation marker instead of the markers used for Fbw7 KO. Only one marker is insufficient, Fbxw7 markers should be used.

Please see response to point 4.

12) “These data suggest FBW7 controls the abundance of TALPID3”: there is no influence on the abundance demonstrated nor accumulation of TALPID3 in double knockout cells as suggested by the abstract.

We tried very hard to detect endogenous TALPID3 in C3H10T1/2 cells using a commercially available antibody from Proteintech, which detected transfected TALPID3, but with no luck. We have shown decreased levels of transfected TALPID3 when co-expressed with the three isoforms of FBW7 (Fig. 7b, third panel from top). We also show that TALPID3 is a direct target of FBW7 in ubiquitinylation experiments (Fig. 7c).

- Minor points:

Introduction:

- I would suggest to replace „Maximal hedgehog activity“ by “adequate hedgehog activity”. Mouse models eg IFt144 suggest that cilia dysfunction can also result in a broadened hedgehog activity pattern, so cilia do not necessarily increase hedgehog activity overall, they are more responsible for properly controlled adequate hedgehog activity.

Text has been modified.

Results:

- Its confusing that Title and large parts of the introduction use FBW7 as a term and at the beginning of the results, Fbw7 is used. I would suggest to use a singular term or add a very short explanation in the introduction.

FBW7 refers to the protein, whereas *Fbxw7* refers to the gene.

- “ However, since *Nde1*-null cells have normal ciliation and even longer primary cilia compared to wild type, we hypothesized that functionality of cilia related Hedgehog effectors might have been compromised.” Again, longer than normal cilia suggest a structural problem which could in turn cause a functional problem.

Please see response to point 7.

Figure 7A co-IP: Why are both proteins myc tagged in this co-IP, authors use an HA version in the figure part on the right as well?

FBW7-myc is a truncated form of FBW7, widely used in functional studies. It contains the common portion of all three isoforms. HA-tagged constructs express full length specific isoforms of FBW7 (α , β and γ). The experiment in Fig. 7a was performed to provide additional supporting evidence for the FBW7-TALPID3 interaction.

Reviewer #3:

The authors seek to identify the role of two components previously identified as regulators of primary cilium formation/function in mesenchymal stem cell differentiation and, ultimately, bone tissue maintenance. In this work, they reveal that a protein network consisting of Fbxw7/Nde1/TALPID3 coordinates primary cilium function and Hh signaling (via the cilium or independently) to encourage osteogenic differentiation. Overall, the manuscript is very well written and organized, the experimental design is clear and appropriate, and the authors' conclusions are supported by their data. Furthermore, the results are novel for the primary cilium and bone cell fields and will likely contribute to further investigation by other researchers. My comments to strengthen the scientific rigor and novelty of the manuscript are outlined below.

Abstract: My only comment is regarding precise language. The authors do a great job of using the full term "primary cilia" at the beginning of most paragraphs in the rest of the manuscript, but the general term "cilia" is exclusively used in the abstract. It is important to distinguish between primary and motile cilia since these structures and research fields can be very different. I recommend clarifying primary cilia in the beginning of the abstract at the very least and thoroughly checking the rest of the document for precise use of these terms.

The text has been rephrased accordingly.

Introduction: Well organized and clearly written. All the necessary terms are defined and an appropriate amount of background information is provided to set up the reader.

Methods: In general, the methods provide enough detail that experiments can be recapitulated but two key descriptions are missing. First, it is insufficient to simply state "followed by the proper ad hoc test" in the statistics section. According to the figure captions, I only see Dunnett's being used so please explicitly state this in the methods and provide a brief rationale for selecting this ad hoc test. Second, there is no description of how cilium length was measured other than to say it was done using ImageJ. This is not a trivial process and the measurements can be drastically influenced by how length is defined (i.e. did the authors measure in 2D or generate 3D z-stacks to visualize cilium length?), so it is critical for the authors to include this information in their methods.

As suggested by reviewer 1, we added more comparisons between groups in Fig. 4. According to Graphpad software, Dunnett's test is more appropriate for comparing every mean with a control mean. Therefore, we changed our *ad hoc* tests in Fig. 4 to Sidak's, which is more appropriate for comparisons of multiple pairs of means and it is not limited to comparisons only with a control mean. Generally, wherever means were

compared only with the control group (e.g. Fig 5) Dunnett's test was used, whereas in the case of comparisons of multiple pairs of means, Sidak's test was used.

We used ImageJ to quantify ciliary length in z-stack images. We have updated the "Methods" section.

Results:

1) Figure 1: What is the justification for only using male mice? It is well known that sex can influence bone phenotype so it is critical for the authors to provide their reasoning for using only one sex. Moreover, if there is no phenotype in female mice this is important information to disclose. Was cortical bone examined in these mice? C57BL6/J mice famously have very dense cortical bone and are known to not display cortical phenotypes, but many bone researchers would be curious to know if a difference exists.

We focused our studies on male mice to avoid possible hormonal effects on bone architecture in female mice.

We could not detect a difference in cortical BV/TV between WT and *Fbxw7KO* mice. These data have been added (please see new Fig. 1g).

2) Figure 2: Consider using the term "cilium incidence and length" rather than "ciliation". This is more precise wording for % of ciliated cells as a whole -- rather than whether an individual cell has a primary cilium -- and more accurately describes your experimental read outs.

Text has been rephrased throughout, when appropriate.

3) Figure 3: This experiment is well designed, I just have a suggestion regarding data presentation. The multiple bar graphs in (c),(d), and (e) could be neatly organized onto a single plot with the columns for each target gene spaced apart and labeled since they all share the same scale and y-axis label. Figure 3 from a recent publication is a good example of what I am trying to explain: <https://www.nature.com/articles/s42003-021-01675-4>. Currently, the bar graphs fill an unnecessary amount of space and the redundant labels/ axes visually take away from the otherwise great data.

We agree with the reviewer and provide an updated version of the graph (please see new Fig. 3c-e).

4) Figure 4: Great, no comments.

5) Figure 5: I like the concept of this experiment but I have several issues with execution and presentation. First, it is already well established that cells require primary cilia to undergo osteogenic differentiation. In lines 169-170, the wording indicates this is a new finding, when really this experiment simply demonstrates that their techniques successfully disrupted primary cilia.

Furthermore, I am confused as to why Ciliobrevin A treatment was introduced. If the authors want to attribute findings to the primary cilium, *Ift88* deletion is the most accepted way to uniquely disrupt the cilium. Ciliobrevin is known to influence non-ciliary pathways (<https://www.nature.com/articles/nature10936>) and can confound interpretation of the data. The *Ift88* sgRNA data alone are sufficient to demonstrate the authors' point so I would consider

moving all Ciliobrevin A data to a Supplementary Figure unless the authors plan to comment on differences between the two approaches.

Second, Figure 5a/c and b/d are redundant: measuring ALP using a fluorescent stain does not tell the reader anything more than measuring it with a colorimetric stain did. Furthermore, in 5c the intensity of the WB band and the intensity of the ALP staining for the double KO do not appear consistent to my eye. I would expect the band to be much darker since the ALP stain coloration is comparable to the control, such as is the case in Figure 6b/c. I would recommend editing out the unnecessary data in this figure (i.e. the Ciliobrevin A treatment and select either a/c or b/d) and adding ICC looking at primary cilia and ALP simultaneously. This image would reveal whether ALP production by a single cell correlates with the existence of and/or length of the primary cilium. This will help demonstrate whether or not Nde1 and Fbxw7 directly influence ALP production via the cilium.

We changed the text to indicate confirmation of previous studies on cilia and osteogenic differentiation. In addition, we have removed the ciliobrevin A data from the manuscript, according to the reviewer's suggestion that *Ift88* deletion would be more specific and sufficient.

We removed images showing ALP fluorescent staining, as these data were redundant with Fig. 5a, as suggested by the reviewer. New Fig. 5c shows quantification of 5b. We attempted to double stain for cilia and ALP in cells undergoing differentiation, but these experiments were technically challenging, as ALP was somewhat diffuse and not suitable for identifying clearly a differentiating cell. Please also note that we used single clones of *Fbxw7*-, *Nde1*-, and *Fbxw7/Nde1*-null cells, where percentages of ciliated cells and ciliary length were uniform among cells in each clone and carefully quantified (new Fig. 4).

6) Figures 6 and 7 are experimentally sound and provide insights into the individual deletions of Nde1 and Fbxw7, but I am left skeptical that TALPID3 explains the partial recovery in the double KO. I think this can be resolved by presenting the bar graphs in e, f, h, i, k, and l differently. Rather than showing the reduction in Hh signaling and ALP activity normalized to either WT or the double KO cells, everything should be normalized to the WT cells. This way we can see if the double KO cell readouts are slightly decreased compared to WT and recapitulate the data (especially the ALP readouts) presented in earlier figures. The way the bar graphs are currently presented, all it shows is that knockdown of TALPID3 reduces Hh signaling and ALP in both cell types. It would further strengthen the authors claims to increase TALPID3 expression and observe whether ALP and Hh signaling are enhanced in WT and double KO cells.

As suggested by the reviewer, we rearranged the data and showed normalization based on WT (please see new Fig. 7i,j). Concerning the overexpression of TALPID3, our model suggests that TALPID3 is accumulated in the doubleKO cells, although we could not determine by how much, due to the lack of a sensitive enough antibody. Therefore to recapitulate this level of increase by overexpression would be challenging, especially in light of recent data that massive overexpression of TALPID3 leads to multinucleation and centrosome amplification¹², that could complicate data interpretation. Our approach to silence TALPID3 in wild type and double mutant cells provided proof-of-principle that it is needed for osteoblast differentiation in these cells. In response to reviewer 1, we also show that it is needed for cilia formation. Therefore, we can safely conclude that is a positive regulator of cilia formation, Hedgehog signaling, and osteoblast differentiation.

We can suggest that its moderate accumulation in double mutant cells, in which cilia have been restored, TALPID3 could directly (activation, nuclear trafficking, etc) or indirectly (via effects of cilia) increase the amount of activated GLI2 available for signaling.

Figure 8: The cartoon is logical based on the authors' conclusions.

Discussion: The authors succinctly summarize their results, provide reasonable speculations based on their data (i.e. enhancement of TALPID3-mediated Hh signaling via deletion of Fbxw7 can overcome the absence of Nde1-mediated Hh signaling), and nicely discuss how their data informs the existing literature and research fields.

References

1. Roessler, E. *et al.* A previously unidentified amino-terminal domain regulates transcriptional activity of wild-type and disease-associated human GLI2. *Hum Mol Genet* **14**, 2181-2188 (2005).
2. Kelley, J.B. & Paschal, B.M. Hyperosmotic stress signaling to the nucleus disrupts the Ran gradient and the production of RanGTP. *Mol Biol Cell* **18**, 4365-4376 (2007).
3. Kim, S. *et al.* Nde1-mediated suppression of ciliogenesis affects cell cycle re-entry. *Nat Cell Biol* **13**, 351-360 (2011).
4. Maskey, D. *et al.* Cell cycle-dependent ubiquitylation and destruction of NDE1 by CDK5-FBW7 regulates ciliary length. *Embo j* **34**, 2424-2440 (2015).
5. Kobayashi, T., Kim, S., Lin, Y.C., Inoue, T. & Dynlacht, B.D. The CP110-interacting proteins Talpid3 and Cep290 play overlapping and distinct roles in cilia assembly. *J Cell Biol* **204**, 215-229 (2014).
6. Hu, H. *et al.* Sequential roles of Hedgehog and Wnt signaling in osteoblast development. *Development* **132**, 49-60 (2005).
7. Kato, M. *et al.* Cbfa1-independent decrease in osteoblast proliferation, osteopenia, and persistent embryonic eye vascularization in mice deficient in Lrp5, a Wnt coreceptor. *J Cell Biol* **157**, 303-314 (2002).
8. Guo, X. *et al.* Wnt/beta-catenin signaling is sufficient and necessary for synovial joint formation. *Genes Dev* **18**, 2404-2417 (2004).
9. Voloshanenko, O. *et al.* β -catenin-independent regulation of Wnt target genes by RoR2 and ATF2/ATF4 in colon cancer cells. *Scientific reports* **8**, 3178 (2018).
10. Person, A.D., Garriock, R.J., Krieg, P.A., Runyan, R.B. & Klewer, S.E. Frzb modulates Wnt-9a-mediated beta-catenin signaling during avian atrioventricular cardiac cushion development. *Dev Biol* **278**, 35-48 (2005).
11. Yoon, J.W. *et al.* p53 modulates the activity of the GLI1 oncogene through interactions with the shared coactivator TAF9. *DNA Repair (Amst)* **34**, 9-17 (2015).
12. Wang, J. *et al.* Talpid3-Mediated Centrosome Integrity Restrains Neural Progenitor Delamination to Sustain Neurogenesis by Stabilizing Adherens Junctions. *Cell reports* **33**, 108495 (2020).

REVIEWERS' COMMENTS:

Reviewer #1 (Remarks to the Author):

The authors have done a good job in addressing my comments.

Reviewer #3 (Remarks to the Author):

The authors did an excellent job of addressing my comments, as well as the comments from the other reviewers. I believe the manuscript is now acceptable for publication.